# Stochastic gradient updates yield deep equilibrium kernels

**Russell Tsuchida**                                                *russell.tsuchida@data61.csiro.au*
*Data61, CSIRO*

**Cheng Soon Ong**                                                *chengsoon.ong@anu.edu.au*
*Data61, CSIRO*
*Australian National University*

**Reviewed on OpenReview:** *https:// openreview. net/ forum? id=p7UTv2hWgM*

## Abstract

Implicit deep learning allows one to compute with implicitly defined features, for example features that solve optimisation problems. We consider the problem of computing with implicitly defined features in a kernel regime. We call such a kernel a deep equilibrium kernel (DEKER). Specialising on a stochastic gradient descent (SGD) update rule applied to features (not weights) in a latent variable model, we find an exact deterministic update rule for the DEKER in a high dimensional limit. This derived update rule resembles previously introduced infinitely wide neural network kernels. To perform our analysis, we describe an alternative parameterisation of the link function of exponential families, a result that may be of independent interest. This new parameterisation allows us to draw new connections between a statistician's inverse link function and a machine learner's activation function. We describe an interesting property of SGD in this high dimensional limit: even though individual iterates are random vectors, inner products of any two iterates are deterministic, and can converge to a unique fixed point as the number of iterates increases. We find that the DEKER empirically outperforms related neural network kernels on a series of benchmarks. [1]

## 1 Kernel methods, deep learning and implicit deep learning

Kernel methods are a classical paradigm for analysing representational capacity, bias, generalisation performance and practical algorithms for nonparametric prediction (Schölkopf et al., 2002). Many classical nonparametric models can be seen as extensions of parametric models (Saunders, 1998; Rasmussen & Williams, 2006, § 2.2) that allow for increased representational capacity while retaining some statistical model-based properties. Examples of model-based qualities may include the smoothness, stationarity or periodicity of the predictor (Duvenaud, 2014, § 2) or the statistical interpretation of the learning procedure (Sollich, 2002; Rasmussen & Williams, § 3), which may be understood by examining the kernel or the loss function (Banerjee et al., 2005, Theorem 4).

Despite early successes of kernel methods, when data is plentiful and/or modelling is hard, over-parameterised and under-regularised deep learning is now seen as the dominant paradigm for practical nonparametric-style prediction (OpenAI et al., 2019; Adiwardana et al., 2020; Rombach et al., 2022). Unlike parametric and classical nonparametric approaches, the architecture and loss functions of many explicit neural networks are driven purely from the perspective of representational power or predictive performance (either empirical (Vaswani et al., 2017) or mathematical (Raghu et al., 2017)) rather than model-based qualities.

A fruitful direction is to analyse deep learning predictors through the reductionist lens of kernel methods through sufficiently well-behaved neural networks in certain large parameter count regimes (Neal, 1995).

---

[1] Code available at `https://github.com/RussellTsuchida/dek.git`.

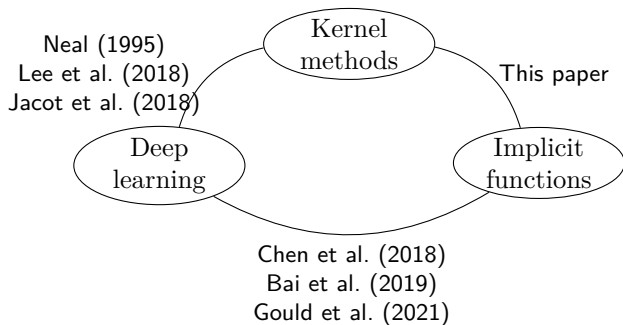

Figure 1: We establish links between kernel methods and implicit functions to design a kernel with corresponding statistical assumptions about a latent variable model.

However, to the best of our knowledge, no current theory describes architectural properties of neural networks in the kernel regime such as choice of activation function, depth and skip connections, in terms of model-based properties. It is desirable to motivate predictive deep learning architectures from a more fundamental, statistical model-based perspective (Rudin, 2019; Efron, 2020) in a kernel regime.

Implicit neural networks are an emerging approach to model-based deep learning. We describe such networks as model-based because the layers are defined and guaranteed to implicitly satisfy the solution to a problem arising from a model that is not only purely predictive, but also conceptually elegant. Implicit networks use solutions to problems as feature representations. For example, deep declarative networks (DDNs) (Gould et al., 2021) solve optimisation problems, deep equilibrium models (DEQs) (Bai et al., 2019) solve fixed point (algebraic) problems and neural ODEs solve differential equations (Chen et al., 2018). Example applications of implicit neural network layers include layers that model optimal transport layers (Campbell et al., 2020; Eisenberger et al., 2022) and layers that perform point estimation of specific statistical models (Tsuchida et al., 2022; Tsuchida & Ong, 2023). Such problems are usually computed numerically via the (approximate) fixed point of an iterative procedure. This leads to the view that implicit layers are themselves a composition of infinitely many functions. Owing to the complexity of deep learning algorithms, theory falls short of explaining the empirically demonstrated successes of both implicit and explicit models. To the best of our knowledge, no general notion of an implicit kernel is currently described in the literature.

We define a new type of kernel called a deep equilibrium kernel. This kernel is defined as the inner product of features, where features are taken to be solutions to a given problem depending on inputs, using a given algorithm, at a given iteration of the algorithm. We focus on the special case where the problem is an optimisation problem for point estimates of a particular latent variable model and the algorithm is stochastic gradient descent. We find that in the limit as the dimensionality of the features goes to infinity, the deep equilibrium kernel at any iterate can be represented as a function of the deep equilibrium kernel at the previous iterate. We describe our contributions in more detail in § 1.1. Our analysis brings together elements of implicit functions, kernel methods, and deep learning (Figure 1).

## 1.1 Our contributions: an implicit kernel and an update rule in kernel space

**Updates in feature space** Solutions to optimisation, fixed point or differential equation problems are in practice most often obtained via a possibly stochastic iterative update procedure. Let $X_1 \in \mathbb{X} \subseteq \mathbb{R}^l$ be an input to the problem and $\psi_{X_1} \in \boldsymbol{\psi} \subseteq \mathbb{R}^m$ be the solution to the problem. Note that $\psi_{X_1}$ is the evaluation of an implicit function of $X_1$. The function is implicit because the mapping is defined through the solution to a problem, rather than an explicit closed-form expression. Let $\psi_{X_1}^{(0)}$ be an initial guess for solution at iterate 0. Suppose there exists some possibly stochastic function $g^{(t)}(\,\cdot\,; X) : \boldsymbol{\psi} \to \boldsymbol{\psi}$ that maps solutions at iterate $t$ to solutions at iterate $t + 1$, so that

$$\psi_X^{(t+1)} = g^{(t)}\big(\psi_X^{(t)}; X\big). \tag{1}$$

We call $\psi_X^{(t)}$ features and $g^{(t)}$ the update rule in feature space. We emphasise that we consider the problem where features are updated, not weight parameters as in some other settings. This is unlike typical settings for the neural tangent kernel (Jacot et al., 2018), where weight and parameter updates are performed for a fixed depth.

**Deep equilibrium kernels**  We find helpful the notion of an implicitly defined kernel, which we call a deep equilibrium kernel (DEKER). This allows us to draw parallels between infinitely wide implicit neural networks and implicitly defined kernel machines. Let $\psi_{X_1}^{(t+1)}$ and $\psi_{X_2}^{(t+1)}$ be two features corresponding to inputs $X_1$ and $X_2$. Recall that $m$ is the dimension of the feature mapping and $t$ is the iteration of the solver. We consider three kernel evaluations in terms of the implicit updates in feature space,

$$\underbrace{\overline{\Psi}_{12}^{(t+1)} \triangleq {\psi_{X_1}^{(t+1)}}^\top \psi_{X_2}^{(t+1)},}_{\text{finite feature DEKER (ffDEKER)}} \qquad \underbrace{\Psi_{12}^{(t+1)} \triangleq \operatorname*{plim}_{m \to \infty} \overline{\Psi}_{12}^{(t+1)},}_{\text{DEKER}} \qquad \text{and} \qquad \underbrace{\Psi_{12} \triangleq \lim_{\tau \to \infty} \Psi_{12}^{(\tau)},}_{\text{limiting DEKER (}\ell\text{DEKER)}} \qquad (2)$$

where defined, where plim denotes convergence in probability. Note the order of the limits. We will similarly write $\Psi_{11}$ and $\Psi_{22}$ to represent evaluations of such DEKERs at $(X_1, X_1)$ and $(X_2, X_2)$ respectively. We write

$$\Psi^{(t+1)} = \begin{pmatrix} \Psi_{11}^{(t+1)} & \Psi_{12}^{(t+1)} \\ \Psi_{21}^{(t+1)} & \Psi_{22}^{(t+1)} \end{pmatrix}$$

for the corresponding $2 \times 2$ PSD matrices (and likewise for the ffDEKER and $\ell$DEKER). The dimensionality of features are allowed to grow to infinity, but only after taking the inner product, resulting in a scalar value for examination. The mathematical construction of $2 \times 2$ matrices suffices for our purposes to analyse algorithms that utilise $N$ examples $X_1, \ldots, X_N$, since every element of an $N \times N$ kernel matrix is an element of a corresponding $2 \times 2$ matrix. Defining such a kernel allows one to build predictive algorithms that operate on kernel matrices instead of feature space, avoiding the necessity of describing, analysing and building algorithms involving infinite dimensional feature spaces.

> **Updates in kernel space**  Let $\mathbb{S}_+^2 = \{K \in \mathbb{R}^{2 \times 2} \mid K = K^\top, K \succeq 0\}$ denote the space of $2 \times 2$ PSD matrices. Our central questions are as follows. Firstly, as $m \to \infty$, does there exist an update that may be performed on $2 \times 2$ PSD DEKER matrices instead of $2m$-dimensional feature space? Secondly, can we write a closed form for the update? Finally, do repeated iterations of the update converge? That is, does there exist a closed-form update rule in kernel space $G(\,\cdot\,; X_1, X_2) : \mathbb{S}_+^2 \to \mathbb{S}_+^2$ such that
>
> $$\Psi^{(t+1)} = G(\Psi^{(t)}; X_1, X_2)\,? \qquad (3)$$
>
> And does $\qquad \Psi = G(\Psi) = \lim_{\tau \to \infty} \underbrace{G \circ \ldots \circ G}_{\tau \text{ compositions}} (\Psi^{(0)}; X_1, X_2)\,? \qquad (4)$

For convenience, we notationally decompose $G$ into components via a function $G$ satisfying for each $ij \in \{11, 22, 12\}$

$$G(\Phi; X_1, X_2) = \begin{pmatrix} G_{11} & G_{12} \\ G_{12} & G_{22} \end{pmatrix}, \qquad \text{where}$$

$$G_{ij} \triangleq G(\Phi_{ii}, \Phi_{jj}, \Phi_{ij}; X_i, X_j) \triangleq \Big( G(\Phi; X_1, X_2) \Big)_{ij}. \qquad (5)$$

Here $\Phi \in \mathbb{S}_+^2$ is a local dummy variable whose sole purpose is to define $G$ and $G$.

**Contributions**  We study an important special case of a DEKER where we answer equation 3 and equation 4 positively, one in which the features are iteratively updated using SGD applied to a latent variable model. The model is *over-parameterised* (the number of parameters grows much faster than the amount of data), but shallow (the *motivation* for the model more closely resembles exponential

family PCA than a deep neural network). The objective to which we apply SGD is an *under-regularised* variant of an expected negative log posterior for point estimates of latent variables. Our main result (stated precisely in § 3) is a constructive proof of the existence of a closed form update rule in kernel space $\mathsf{G}$.

Surprisingly, despite the feature space of the DEKER having seemingly no direct relation with deep learning predictors, deep learning structures emerge as part of our analysis. Our DEKER is a kernel which satisfies a fixed point equation, constructed via a latent variable model optimisation problem solved using SGD. Our DEKER may be understood (but is not constructed) as an infinitely wide DEQ whose iterates are computed with stochastic mapping that is resampled at every iterate rather than as a deterministic fixed point iteration. Further, the kernel iterates of our DEKER resemble previously introduced NNKs and NTKs and may be computed with deterministic fixed point solvers.

Informally, our main result (Theorem 4) states that when the features $\psi_{X_1}^{(t)}$ and $\psi_{X_2}^{(t)}$ are point estimates obtained by SGD, we can construct a deterministic update rule, $\mathsf{G}$, for the DEKER, $\Psi^{(t+1)}$. Further, Corollary 5 says that repeated applications of $\mathsf{G}$ converge to a fixed point, the $\ell$DEKER. We further quantify the degree to which the $\ell$DEKER is an invariant of SGD (i.e. does not change as iterates of SGD increase) when treated as an ffDEKER (Theorem 7).

## 2    Background

Our analysis requires combining fixed point theory (optimisation), the exponential family (statistics), and neural network kernels (machine learning). We briefly describe elements of these topics here.

### 2.1    Notation

Numerical subscripts are used to extract (groups of) indices of a vector or matrix. Parenthesised superscripts indicate a layer or iteration of a naive fixed point solver, both of which turn out to be the same in our constructions, as is consistent with other DEQ works. We index objects by iteration by superscript $(t)$, so that $\psi^{(t)}$ represents a feature in the $t$th iteration. Sans serif fonts are used to denote matrices, and serif fonts are used to denote vectors (so $\mathsf{W}$ is a matrix and $W$ is a vector). We use $\phi$ and $\Phi$ for arbitrary vectors and inner products that are not necessarily obtained by iterations of SGD. We will use $\psi$ and $\Psi$ for feature mappings and inner products of feature mappings that are obtained by iterations of SGD.

We assume that we are given access to a dataset $\mathsf{X} \in \mathbb{R}^{N \times l}$ of $N$ examples of datapoints $X_i \in \mathbb{X} \subseteq \mathbb{R}^l$. We denote by $X_1$ and $X_2$ any two elements of this dataset.

There are two types of function signatures we associate with PSD kernels. The first is for a usual PSD kernel $k : \mathbb{X} \times \mathbb{X} \to \mathbb{R}$, so that an evaluation is written $k(X, X')$ for any two $X, X' \in \mathbb{X}$. We call this form a $k$-form kernel. The second is for a PSD kernel whose evaluation depends on $\phi_1, \phi_2 \in \boldsymbol{\psi}$ only through evaluations $\Phi_{11} = \langle \phi_1, \phi_1 \rangle, \Phi_{12} = \langle \phi_1, \phi_2 \rangle, \Phi_{22} = \langle \phi_2, \phi_2 \rangle$ of some suitably defined inner product $\langle \cdot, \cdot \rangle : \boldsymbol{\psi} \times \boldsymbol{\psi} \to \boldsymbol{\Psi}$. We represent such a kernel through $\kappa : \boldsymbol{\Psi}^3 \to \mathbb{R}$ with evaluations $\kappa\big(\langle \phi_1, \phi_1 \rangle, \langle \phi_2, \phi_2 \rangle, \langle \phi_1, \phi_2 \rangle\big)$. An example of this second form is the NNK (equation 11). We call this form a $\kappa$-form kernel.

Our notation is summarised in Table 3 in Appendix A.

### 2.2    Fixed points and infinite compositions

Let $f : \mathbb{F} \to \mathbb{F}$ for some set $\mathbb{F}$ equipped with a norm $\| \cdot \|$ and norm-induced metric. A fixed point of $f$ is any $Z^* \in \mathbb{F}$ satisfying $Z^* = f(Z^*)$. Banach's fixed point theorem (BFPT) (Goebel & Kirk, 1990, Theorem 2.1) gives sufficient conditions for the existence and uniqueness of such a fixed point.

**Theorem 1** (BFPT). *Let $(\mathbb{F}, \| \cdot \|)$ be a non-empty complete normed space. A mapping $f : \mathbb{F} \to \mathbb{F}$ is called a contraction mapping if there exists some $q \in [0, 1)$ such that $\|f(Z) - f(Z')\| \leq q\|Z - Z'\|$ for*

*every $Z, Z' \in \mathbb{F}$. Every contraction mapping $f$ admits a unique fixed point $Z^* \in \mathbb{F}$. Furthermore, for any initial element $Z^{(1)} \in \mathbb{F}$, the sequence $Z^{(t+1)} = f(Z^{(t)})$ for $t \geq 1$ converges to $Z^*$ as $t \to \infty$.*

It is worth noting that BFPT not only provides a mathematical condition for well-posedness, but also describes an algorithm for approximating fixed points of contraction mappings. We call this algorithm the *naive fixed point solver*, which simply involves applying a $\tau$-fold composition of $f$ to some starting value $Z^{(1)}$, with a linear rate of convergence immediate from the definition of contraction mapping, i.e. $\|Z^* - Z^{(t+1)}\| \leq \frac{q^t}{1-q} \|Z^{(2)} - Z^{(1)}\|$. Other solvers for fixed point problems are available, many of which are approximate Newton methods for root finding (Kelley, 1995).

Deep equilibrium models (DEQs) (Bai et al., 2019) are neural network predictors constructed of parameterised layers that output the solution to fixed point equations $Z^* = f_{\mathsf{U}}(Z^*)$, where $\mathsf{U}$ is a general parameter object. These layers draw upon earlier works on recurrent backpropagation (Pineda, 1987; Almeida, 1990), leveraging the modern machinery of deep learning architectures, optimisers and heuristics. The unsupervised learning problem for a DEQ, an example of which is considered by (Tsuchida & Ong, 2023), is

$$\underbrace{\min_{\mathsf{U}} \sum_{i=1}^{N} L\big(X_i, Z_i^*, \mathsf{U}\big)}_{\text{Empirical risk minimisation for parameters } \mathsf{U}} \qquad \text{subject to} \qquad \underbrace{Z_i^* = f_{\mathsf{U}}(Z_i^*, X_i),}_{\text{Fixed point solution for DEQ predictions } Z_i^*}$$

where $L$ is some loss function, $\mathsf{U}$ is a parameter object, and $\{X_i\}_{i=1}^{N}$ is a collection of input examples (a supervised setting might also involve a set of output examples). Derivatives $\frac{\partial Z_i^*}{\partial \mathsf{U}}$ of outputs of these layers with respect to their parameters $\mathsf{U}$ can be computed without backpropagating through the iterates of the fixed point solver using the implicit function theorem (Bai et al., 2019). This allows first-order stochastic gradient methods that are popular with explicit deep learning architectures to be applied to DEQs.

In general, it is not guaranteed that a function necessarily admits a unique fixed point; various works discuss dealing with multiple fixed points or ensuring or encouraging that exactly or at least one fixed point exists (Winston & Kolter, 2020; Revay et al., 2020; El Ghaoui et al., 2021). Interestingly, if a single DEQ layer involves finding the fixed point of a contraction mapping, by Theorem 1 the output computed by a DEQ has the interpretation of an infinitely deep neural network with shared parameters in each layer. More generally, mappings computed by the naive fixed point solver have interpretations as very deep neural networks with shared parameters in each layer. Since zeros of the gradient of sufficiently well behaved objectives are stationary points of the objectives, DEQ layers share a connection with optimisation-based implicit layers (Gould et al., 2021), as explored in various works (Revay et al., 2020; Xie et al., 2021; Tsuchida et al., 2022; Riccio et al., 2022; Tsuchida & Ong, 2023). Our current investigation concerns a connection more specific than optimisation, since it considers the special case of applying SGD.

### 2.3 Exponential families

**Exponential families** The feature mappings that we use to build our kernel are estimates obtained using SGD applied to certain variants of exponential family likelihoods and Gaussian priors. We now define *minimal and regular exponential families in canonical form*. Let $h$ be a probability density (mass) function supported on data space $\mathbb{Y} \subseteq \mathbb{R}$. Let $T : \mathbb{Y} \to \mathbb{R}$ be a function called the *sufficient statistic*. Given some *canonical parameter* $\eta$ belonging to an open set $\mathbb{H} \subseteq \mathbb{R}$, we may construct a probability density (mass) function by normalising the nonnegative function $h(\cdot) \exp\big(T(\cdot)\eta\big)$. The normalising constant is called the partition function, and its strictly convex and infinitely differentiable logarithm $A$ is called the *log partition function* (Wainwright et al., 2008, Proposition 3.1). We write

$$p(y \mid \eta) = h(y) \exp\big(T(y)\eta - A(\eta)\big), \quad A(\eta) = \log \int_{\mathbb{Y}} h(y) \exp\big(T(y)\eta\big) \, dy$$

for the evaluation of a probability density (mass) function of an exponential family. The log partition function $A$ acts as a cumulant generating function for the conditional distribution of the sufficient statistic $T$. In particular the expected value of the sufficient statistic (often called the *expectation parameter* (Nielsen & Garcia, 2009)) is the gradient of the log partition function $A$. That is,

$$\mathbb{E}\big[T(y) \mid \eta\big] = A'(\eta). \tag{6}$$

We consider factorised exponential families in the following sense. Let $y_1, \dots y_d$ be distributed according to the same exponential family and define data vector $Y = (y_1, \dots, y_d)^\top$ and canonical parameter vector $H = (\eta_1, \dots, \eta_d)$. Then the joint distribution of data $Y$ conditioned on canonical parameters $H$ is the product of the individual elements

$$p(Y \mid H) = \prod_{i=1}^{d} p(y_i \mid \eta_i) = \Big( \prod_{r=1}^{d} h(y_r) \Big) \exp \big( T(Y)^\top H - A(H)^\top \mathbf{1} \big), \tag{7}$$

where we write $T(Y) = \big( T(y_1), \dots, T(y_d) \big)^\top$, $A(H) = \big( A(\eta_1), \dots, A(\eta_d) \big)^\top$ and $\mathbf{1} = (1, \dots, 1)^\top$.

**Link functions and canonical link functions**   Exponential families are used in generalised linear models (GLMs) (McCullagh & Nelder, 1989). In GLMs, the conditional expectation equation 6 of an exponential family is set to be the result of applying an (invertible) *inverse link function* $s^{-1}$ to the result of a linear transformation of features $\phi \in \mathbb{R}^m$ (classically called parameters). That is, for some linear basis $\mathsf{V} \in \mathbb{R}^{d \times m}$ (classically called covariates),

$$A'(H) = \mathbb{E}\big[T(Y) \mid H\big] = s^{-1}(\mathsf{V}\phi). \tag{8}$$

The conditional expectation is then mapped to the canonical parameter $H$ through $H = (A')^{-1} \circ s^{-1}(\mathsf{V}\phi)$, noting that $A'$ is invertible because $A$ is strictly convex. In the case where $s^{-1}$ is chosen to be $A'$, $s \equiv (A')^{-1}$ is called the *canonical link function*, and we observe from equation 8 that the canonical parameter and conditional expectation satisfy

$$H = \mathsf{V}\phi, \qquad \mathbb{E}[T(Y) \mid \mathsf{V}\phi] = A'(\mathsf{V}\phi) = s^{-1}(\mathsf{V}\phi). \tag{9}$$

There are two main and sometimes conflicting reasons why one might be interested in using a non-canonical link function. The first is computational; if the link function were canonical, for some distributions such as Gamma or exponential one would need a constrained optimisation method over the open set $\mathbb{H}$ instead of $\mathbb{R}$. If $s^{-1}$ were allowed to be non-canonical — that is, we are free to choose $s^{-1}$ different from $A'$ — we could map the conditional expectation to the appropriate constraint set and unconstrained optimisation procedures could be applied. In a Bayesian context, sampling from the posterior over $\phi$ can be made easier by convenient choices of $s$. For example, the probit model admits an efficient Gibb's sampler for the posterior (Albert & Chib, 1993). The second, and arguably more important consideration is modelling; we might have reason to suspect that the conditional expectation is constrained. For example, if the observations should have a positive expectation, the power family of link functions might be used (McCullagh & Nelder, 1989, equation 2.9a). Alternative link functions can lead to exploiting particular properties of interest; for example, Wiemann et al. (2021) use the softplus function for positive conditional expectations to exploit its identity-like behaviour at large positive values. In weighing up the possibly conflicting aims of computational convenience and modelling suitability, we highlight the view of Efron & Hastie (2021, page 68); while classical exponential families and link functions may lead to closed-form expressions, modern computer technology allows us more flexible models.

**Point estimation**   When using a canonical inverse link function $s \equiv A'$, the negative logarithm of the likelihood (equation 7) is strictly convex in $H$, since $A$ is strictly convex and linear functions are convex. If $H$ is chosen to be $H = \mathsf{V}\phi$, this translates to convexity in $\phi$, and maximum likelihood estimates can be computed using first or (more typically, in a classical setting) second order optimisation methods. When $s^{-1}$ is not a canonical inverse link function, convexity does not necessarily hold. Nevertheless,

local estimates are practically useful, so pre-implemented link functions and the option to implement custom link functions is available in a number of software frameworks including `R` (R Core Team, 2021, `family`) and `Stata` (Hardin & Hilbe, 2018, `glm`).

## 2.4 Kernels arising from neural networks

Our main result describes the DEKER update rule as a composite function involving evaluations of kernels of a particular form. These are kernels that are constructed from neural network models. In this section, we describe such kernels.

The neural network kernel was first investigated as the covariance function of a certain neural network with random parameters and a single hidden fully connected layer (Neal, 1995). Under mild conditions, as the width of the hidden layer goes to infinity the neural network converges to a Gaussian process. This analysis has since been extended to handle multiple layers (Matthews et al., 2018; Lee et al., 2018), other layer types including convolutional layers (Mairal et al., 2014; Garriga-Alonso et al., 2018; Novak et al., 2019; Yang, 2019a;b), and training under gradient flow via the neural tangent kernel (NTK) (Jacot et al., 2018) . Since our motivation is better described in terms of inner products of the features, we favour the view of the neural network kernel as an inner product in an infinitely wide hidden layer rather than a covariance function of a Gaussian process. We note that connections between Bayesian Gaussian processes and kernel methods exist (Kanagawa et al., 2018) and apply to some but not all infinitely wide neural networks.

**Neural network kernel, single hidden layer**   Let $\mathsf{W}^{(1)} \in \mathbb{R}^{d \times n}$ be the weights of a fully connected hidden layer with activation function $\zeta$ defined over the reals. Suppose each entry of $\mathsf{W}^{(1)}$ is i.i.d. with distribution $\mathcal{N}(0,1)$[2]. Given an input feature $\phi_1 \in \mathbb{R}^{n \times 1}$ (we take the convention that vectors are column vectors), the signal in the hidden layer is $h^{(1)} \triangleq \zeta(\mathsf{W}^{(1)} \phi_1)$[3]. Here and throughout the paper the symbol $\triangleq$ means that the object on the left hand side is defined to be the expression on the right hand side. By a strong law of large numbers, a suitably normalised inner product in the hidden layer converges almost surely as $d \to \infty$ to an expectation,

$$\frac{1}{d} h_1^{(1)\top} h_2^{(1)} = \frac{1}{d} \zeta(\mathsf{W}^{(1)} \phi_1)^\top \zeta(\mathsf{W}^{(1)} \phi_2) \overset{a.s.}{\to} \mathbb{E}_W \big[ \zeta(W^\top \phi_1) \zeta(W^\top \phi_2) \big],$$

assuming the right hand side is finite, since the inner product is a sum of i.i.d. random variables. Here $W^\top \in \mathbb{R}^{1 \times m}$ is a vector with i.i.d. entries drawn from $\mathcal{N}(0,1)$. We define

$$k_\zeta(\phi_1, \phi_2) \triangleq \mathbb{E}_W \big[ \zeta(W^\top \phi_1) \zeta(W^\top \phi_2) \big], \tag{10}$$

and call $k_\zeta$ a single hidden layer neural network kernel (NNK) with activation function $\zeta$. The PSD kernel $k_\zeta$ uniquely defines an RKHS by the Moore–Aronszajn theorem. Closed-form expressions of $k_\zeta$ for different $\zeta$ are available (Williams, 1997; Le Roux & Bengio, 2007; Cho & Saul, 2009; Tsuchida et al., 2018; Pearce et al., 2019; Tsuchida, 2020; Meronen et al., 2020; Tsuchida et al., 2021; Han et al., 2022).

Define $(\chi_1, \chi_2)^\top \triangleq \big( W^\top \phi_1, W^\top \phi_2 \big)^\top$, which is a zero mean bivariate Gaussian with a covariance matrix $\mathbf{\Sigma}^{(1)}$. Note that $k_\zeta(\phi_1, \phi_2)$ depends on the input features $\phi_1$ and $\phi_2$ only through the covariance matrix $\mathbf{\Sigma}^{(1)}$. It is helpful to explicate this dependence structure through a special notation. We have that equation 10 is equal to

$$\kappa_\zeta\big(\Sigma_{11}^{(1)}, \Sigma_{22}^{(1)}, \Sigma_{12}^{(1)}\big) \triangleq k_\zeta(\phi_1, \phi_2) = \mathbb{E}_{(\chi_1, \chi_2)^\top \sim \mathcal{N}(\mathbf{0}, \mathbf{\Sigma}^{(1)})} \big[ \zeta(\chi_1) \zeta(\chi_2) \big], \quad \mathbf{\Sigma}^{(1)} \triangleq \begin{pmatrix} \phi_1^\top \phi_1 & \phi_1^\top \phi_2 \\ \phi_2^\top \phi_1 & \phi_2^\top \phi_2 \end{pmatrix}. \tag{11}$$

With an abuse of terminology, we refer to both $k_\zeta$ and $\kappa_\zeta$ as PSD single hidden layer NNKs. For a more detailed description of the NNK, see Appendix C.

---

[2] The effect of non-unit weight variance may be obtained by scaling all inputs $\phi_1$ by a hyperparameter. Similarly, arbitrary covariance structures inside rows of $\mathsf{W}^{(1)}$ can be reflected as linear transformations of all inputs $\phi_1$.

[3] The effect of zero mean Gaussian biases may be obtained by augmenting inputs with an additional coordinate. The magnitude of this coordinate is equivalent to the quotient of the standard deviation of the weights to the biases.

**Neural network kernel, $\tau$ hidden layers** One may compose equation 11 multiple times by applying a sequence of kernels to a 3-dimensional state represented by a $2 \times 2$ PSD matrix $\mathbf{\Sigma}^{(t)}$, in place of the infinitely wide signals. This 3-dimensional state represents the two squared norms and inner product in each hidden layer. For $t = 1, \dots, \tau$ and $ij \in \{11, 22, 12\}$,

$$\Sigma_{ij}^{(t+1)} \triangleq \mathbb{E}_{(\chi_i, \chi_j)^\top \sim \mathcal{N}(\mathbf{0}, \mathbf{\Sigma}^{(t)})} \left[ \zeta(\chi_i) \zeta(\chi_j) \right] = \kappa_\zeta \left( \Sigma_{ii}^{(t)}, \Sigma_{jj}^{(t)}, \Sigma_{ij}^{(t)} \right), \tag{12}$$

where $\Sigma_{ij}^{(t)}$ denotes the $ij$th element of $\mathbf{\Sigma}^{(t)}$. This iteration appears in deep infinitely wide NNKs (Matthews et al., 2018; Lee et al., 2018). We will refer to this kernel as the $\tau$ layer NNK. Evaluations $\Sigma_{12}^{(\tau+1)}$ of the PSD kernel are determined entirely by the activation function $\zeta$, and uniquely define an RKHS.

**Neural tangent kernel, $\tau$ hidden layers** This kernel describes the limiting behaviour of randomly initialised neural networks that are trained under gradient flow (Jacot et al., 2018). The kernel iterations are similar to equation 12, but also contain components involving the derivative $\dot\zeta$ of $\zeta$. Let $\odot$ denote elementwise product. In addition to the iteration equation 12, let $\Theta^{(1)} = \mathbf{\Sigma}^{(1)}$ and define

$$\Theta^{(t+1)} \triangleq \Theta^{(t)} \odot \dot{\Sigma}^{(t+1)} + \Sigma^{(t+1)}, \quad \text{where} \tag{13}$$

$$\dot{\Sigma}_{ij}^{(t+1)} \triangleq \mathbb{E}_{(\chi_i, \chi_j)^\top \sim \mathcal{N}(\mathbf{0}, \mathbf{\Sigma}^{(t)})} \left[ \dot\zeta(\chi_i) \dot\zeta(\chi_j) \right] = \kappa_{\dot\zeta} \left( \Sigma_{ii}^{(t)}, \Sigma_{jj}^{(t)}, \Sigma_{ij}^{(t)} \right)$$

to obtain the evaluation of the PSD NTK in the last iteration $\Theta_{12}^{(\tau+1)}$. Once again, the kernel is determined entirely by the $\zeta$, and uniquely defines an RKHS. The notion of an NTK for DEQ models has been explored (Feng & Kolter, 2021), which results in a kernel that is in some sense a composition of many layers of kernels. In contrast, our current investigation is about building a compositional kernel from a latent variable model, rather than a neural network model.

While the DEKER we will derive shares only superficial similarities with the NTK, special cases of the DEKER can recover NNKs. We further discuss this at the end of § 3.

## 3 Main results

Our results are most clearly described in terms of an alternative parameterisation of exponential families and link functions, which are perhaps of independent interest. We first describe this alternative parameterisation in § 3.1, before moving onto the setup for our main analysis in § 3.2. We then in § 3.3 provide a special case (Corollary 3) of our main and most general result (Theorem 4) in § 3.4. Finally, quantification of error between the DEKER and ffDEKER is described in § 3.5. We give examples of our resulting updates in Appendix G.2.

### 3.1 An alternative view of link functions in exponential families

Instead of computing via the conditional expectation resulting from the application of an inverse link function equation 8, we follow Tsuchida & Ong (2023) and learn the canonical parameter via a nonlinearity $H = R(\mathsf{V}\phi)$, for some once-differentiable $R : \mathbb{R} \to \mathbb{H}$ called the *canonical nonlinearity*. This means that the conditional likelihood equation 7 is now

$$p(Y \mid \mathsf{V}, \phi) = \prod_{i=1}^{d} p(y_i \mid \eta_i) = \left( \prod_{r=1}^{d} h(y_r) \right) \exp \left( T(Y)^\top R(\mathsf{V}\phi) - A\big( R(\mathsf{V}\phi) \big)^\top \mathbf{1} \right). \tag{14}$$

Such a parameterisation is rich enough to recover the (non-canonical) inverse link function view of the statistician (see Proposition 2). It can therefore be considered to be a change of notation, placing emphasis on the canonical nonlinearity $R$ instead of the inverse link function $s^{-1}$. In our setting, one advantage of such a notation is that it avoids more complicated function compositions involving inverses and derivatives. For example, instead of writing $A \circ (A')^{-1} \circ s^{-1}(\mathsf{V}\phi)$ we may write $A \circ R(\mathsf{V}\phi)$. The value of these simple compositions become more evident in Proposition 2.

| Exponential family | $A(\eta)$ | $s^{-1}(a)$ | $R(a)$ | $\rho(a)$ | $\sigma(a)$ |
|---|---|---|---|---|---|
| Gaussian | $\eta^2/2$ | $s^{-1}(a)$ | $s^{-1}(a)$ | $(s^{-1})'(a)$ | $s^{-1}(a)\,(s^{-1})'(a)$ |
| Gaussian | $\eta^2/2$ | $a$ | $a$ | $1$ | $a$ |
| Gaussian | $\eta^2/2$ | $\mathrm{erf}(a/\sqrt{2})$ | $\mathrm{erf}(a/\sqrt{2})$ | $2p(a)$ | $\mathrm{erf}(a/\sqrt{2})2p(a)$ |
| Gaussian | $\eta^2/2$ | $\mathrm{ReLU}(a)$ | $\mathrm{ReLU}(a)$ | $\mathrm{u}(a)$ | $\mathrm{ReLU}(a)$ |
| Poisson | $\exp(\eta)$ | $s^{-1}(a)$ | $\log s^{-1}(a)$ | $\frac{(s^{-1})'(a)}{s^{-1}(a)}$ | $(s^{-1})'(a)$ |
| Poisson | $\exp(\eta)$ | $\exp(a)$ | $a$ | $1$ | $\exp(a)$ |
| Poisson | $\exp(\eta)$ | $\log(1+\exp a)$ | $\log\log(1+\exp a)$ | $\frac{\exp(a)}{(1+\exp a)\log(1+\exp a)}$ | $\exp(a)/\big(1+\exp(a)\big)$ |
| Bernoulli | $\log(1+\exp(\eta))$ | $s^{-1}(a)$ | | $\frac{(s^{-1})'(a)}{s^{-1}(a)\big(1-s^{-1}(a)\big)}$ | $\frac{(s^{-1})'(a)}{1-s^{-1}(a)}$ |
| Bernoulli | $\log(1+\exp(\eta))$ | $\exp(a)/\big(1+\exp(a)\big)$ | $a$ | $1$ | $\exp(a)/\big(1+\exp(a)\big)$ |
| Bernoulli | $\log(1+\exp(\eta))$ | $P(a)$ | $\log\left(\frac{P(a)}{1-P(a)}\right)$ | $\frac{p(a)}{P(a)P(-a)}$ | $\frac{p(a)}{P(-a)}$ |

Table 1: These examples are obtained by plugging the desired log partition function $A$ and inverse link function $s^{-1}$ into expressions equation 24, equation 25 and equation 26. Canonical link functions are shown in blue. General inverse link function settings are shown in red. Here $P$ and $p$ respectively denote the cdf and pdf of the univariate standard Gaussian and erf denotes the error function.

**Nonlinearities and activation functions**  The derivatives of the log likelihood equation 14 (a *score function*) play a central role in numerical procedures associated with estimation. In our setting, such derivatives involve terms derived from $A$ and $R$. These terms are expressed in terms of functions we call *factor activations* $\rho(a) \triangleq R'(a)$ and *chain activations* $\sigma(a) \triangleq (A \circ R)'(a)$. The following identities show how one may map between choices of $(A, s)$ and choices of $(A, R)$, and additionally how these induce activation functions $\sigma$ and $\rho$ which appear in gradient-based optimisers and our later derivations. Note that we may choose the inverse link function $s^{-1}$ to be non-canonical (not $A'$).

**Proposition 2.** *Consider a regular and minimal exponential family with log partition function $A$ : $\mathbb{H} \to \mathbb{R}$. Suppose the conditional expectation belongs to a set $\mathbb{A}'$, that is, $A'(\eta) \in \mathbb{A}'$ for all canonical parameters $\eta \in \mathbb{H}$. Let $s^{-1} : \mathbb{B} \to \mathbb{A}'$ be an inverse link function, for some $\mathbb{B} \subseteq \mathbb{R}$. That is, for every $\eta \in \mathbb{H}$ there exists some $a \in \mathbb{B}$ such that $A'(\eta) = s^{-1}(a)$. Then equivalently, $\eta = R(a)$, where $R : \mathbb{B} \to \mathbb{H}$ is defined by $R(a) \triangleq \big((A')^{-1} \circ s^{-1}\big)(a)$. Furthermore,*

$$\rho(a) \triangleq R'(a) = \frac{(s^{-1})'(a)}{A'' \circ (A')^{-1} \circ s^{-1}(a)} \qquad and \qquad \sigma(a) \triangleq (A \circ R)'(a) = \frac{s^{-1}(a)\,(s^{-1})'(a)}{A'' \circ (A')^{-1} \circ s^{-1}(a)}.$$

The proof is given in Appendix B. In practice we will take $\mathbb{B} = \mathbb{R}$. We observe that $R$ is the identity if and only if $s$ is a canonical link function (which is to say that $s^{-1}(a) = A'(a)$). For the special case of a Gaussian with known variance, $A'$ is the identity and $R$ is the inverse link function $s^{-1}$. Further cases are listed in Table 1.

Nonlinear parameterisation framed in terms of $R$ instead of $s^{-1}$ are often used (McCullagh & Nelder, 1989, Chapter 11.4 and references therein), but their general relationship to $s^{-1}$ does not appear to be discussed. As our setting is equivalent to using an arbitrary link function, we inherit the motivation of using a non-identity $R$ from the motivation for using a non-canonical link function. We also inherit the usual difficulties in estimation and sample complexity due to using not necessarily canonical link functions.

Recall that the choice of $(A, s)$ should be informed by both modelling and numerical convenience (sampling, optimisation) considerations. Motivated by neural network kernels, we find a different set of $(A, s)$ pairs convenient to work with compared with the generalised linear model setting. Convenience here translates to being able to compute certain Gaussian integrals of the form equation 11 in closed form. For example, we find it easy to work with a Gaussian with non-negative conditional expectation, parameterised by $A(\eta) = \eta^2/2$ and $s^{-1}(a) = \mathrm{ReLU}(a)$, where ReLU is the popular rectified linear unit (Efron & Hastie, 2021, page 362). Another convenient setting is a Gaussian likelihood and probit inverse link function (in contrast with the often seen Bernoulli and probit inverse link function). Our theory holds for general $(A, s)$ pairs, but its practical efficiency is contingent upon the existence of efficient numerical routines for computing the integral equation 11. Such numerical routines in the

absence of closed-forms are available in other works (Zandieh et al., 2021; Han et al., 2022), but we do not study their application here.

## 3.2 Setup

**Stochastic gradient descent** We apply SGD (Robbins & Monro, 1951; Wright & Recht, 2022, Chapter 5) to a minimisation objective $\mathcal{L}(\phi; X) = \mathbb{E}_{\mathsf{V}} L(\phi; X, \mathsf{V})$ with decision variable $\phi$, input $X$ and random linear transformation $\mathsf{V}$. The exact minimisation objective relates to a continuous latent variable model, and is described shortly. Given two inputs $X_1$ and $X_2$, the $t + 1$th iterates are

$$\psi_{X_1}^{(t+1)} = \psi_{X_1}^{(t)} - \alpha^{(t)} \frac{\partial}{\partial \psi_{X_1}^{(t)}} L\big(\psi_{X_1}^{(t)}; X_1, \mathsf{V}^{(t)}\big) \quad \text{and} \quad \psi_{X_2}^{(t+1)} = \psi_{X_2}^{(t)} - \alpha^{(t)} \frac{\partial}{\partial \psi_{X_2}^{(t)}} L\big(\psi_{X_2}^{(t)}; X_2, \mathsf{V}^{(t)}\big) \quad (15)$$

with initial features $\psi_{X_1}^{(0)}$ and $\psi_{X_2}^{(0)}$, a sequence $\{\alpha^{(t)}\}_t$ of step sizes, and a sequence of $\{\mathsf{V}^{(t)}\}_t$ of iid samples of $\mathsf{V}$. We use the features $\psi_{X_1}^{(t)}$ and $\psi_{X_2}^{(t)}$ in equation 15 to define the ffDEKER, DEKER and $\ell$DEKER via equation 2. We stress that we are updating features, not weight parameters.

**Continuous latent variable model** We work with a data generating process which is a slight nonlinear generalisation (Tsuchida & Ong, 2023) of exponential family PCA (Collins et al., 2001), allowing for nonlinear $R$ (or equivalently, non-canonical link functions as in Proposition 2). This model describes data $Y$ as being drawn from an exponential family distribution with a canonical parameter that is a function of a latent $\phi$. The number of conditionally independent observations of exponential family distributed random variables is $d$, and the dimensionality of the latent is $m$.

More concretely, let $X \in \mathbb{X} \subseteq \mathbb{R}^l$ be an input and suppose that data $Y = \Gamma(X)$ follows a factorised exponential family equation 7 for some realisation of a random mapping $\Gamma : \mathbb{X} \to \mathbb{Y}^d$. Let $R : \mathbb{R} \to \mathbb{H}$ be a once-differentiable function. Choose the canonical parameter $H = R(\mathsf{V}\phi)$ to be the composition of $R$ and a linear transformation $\mathsf{V}$ of a latent input variable $\phi \in \mathbf{\Psi} = \mathbb{R}^m$. The linear transformation $\mathsf{V}$ form the model parameters. Place an i.i.d. $\mathcal{N}(0, 1)$ prior over each entry of $\mathsf{V} \in \mathbb{R}^{d \times m}$, independent of $\Gamma$. Place an i.i.d. $\mathcal{N}(0, \lambda^{-1}/m)$ prior over $\phi$. This results in a pre-nonlinearity parameter $\mathsf{V}\phi$ having components with variance which stays constant in $d$ and $m$. For some constant $C$ not depending on $\phi$, we have

$$-\log p\big(\phi \mid \Gamma(X), \mathsf{V}\big) = -\bigg( \underbrace{\log p\big(\Gamma(X) \mid R(\mathsf{V}\phi)\big)}_{\text{Log likelihood}} - \underbrace{m\frac{\lambda}{2}\|\phi\|^2}_{\text{-Log prior}} \bigg) + C.$$

As discussed in Proposition 2, the derivative of the negative log-posterior with log-partition function $A$, which appears in our later optimisation procedure, induces two functions $\rho(a) \triangleq R'(a)$ and $\sigma(a) \triangleq (A \circ R)'(a)$ which we call factor activations and chain activations respectively.

**Objective function** The expected negative log posterior

$$\overline{\mathcal{L}}(\phi; X) \triangleq \mathbb{E}_{\mathsf{V}}\overline{L}(\phi; X, \mathsf{V}), \quad \text{where} \quad \overline{L}(\phi; X, \mathsf{V}) \triangleq \frac{1}{d}\Big( -\log p\big(\Gamma(X) \mid R(\mathsf{V}\phi)\big) + m\frac{\lambda}{2}\|\phi\|^2 \Big), \quad (16)$$

is a commonly used minimisation objective to find point estimates of $\phi$. See Appendix H.1 for a discussion on this objective. The division by $d$ is introduced to account for the natural numerical scaling of the likelihood term, which is a sum of $d$ parts. Following recent deep learning trends, we consider an *over-parameterised* and *under-regularised* variant

$$\mathcal{L}(\phi; X) \triangleq \mathbb{E}_{\mathsf{V}} L\big(\phi; X, \mathsf{V}\big), \quad \text{where} \quad L\big(\phi; X, \mathsf{V}\big) \triangleq \frac{1}{d}\Big( -\log p\big(\Gamma(X) \mid R(\mathsf{V}\phi)\big) + \sqrt{md}\frac{\lambda}{2}\|\phi\|^2 \Big), \quad (17)$$

where $d < m$. We call the setting over-parameterised because there are more parameters than datapoints ($m > d$) and under-regularised because the regularisation strength $\sqrt{md}\frac{\lambda}{2}$ is less than what it would be under a typical Gaussian prior, $m\frac{\lambda}{2}$. This expected negative log posterior may be obtained by choosing an overly broad i.i.d. prior $\mathcal{N}\big(0, \lambda^{-1}/\sqrt{md}\big)$ over $\phi$. We will take $d$ to be a well-behaved function of $m$ such that $d \to \infty$ as $m \to \infty$ (see Assumption 1).

**Assumptions** We now describe two assumptions common to both settings. Our first assumption says that the dimensionality $m$ of the feature space should become larger much faster than the dimensionality $d$ of the number of conditionally independent observations in the exponential family.

**Assumption 1.** *Consider any limit path* $(m, d) \to (\infty, \infty)$ *such that* $\lim\limits_{m \to \infty} \frac{d}{m} = 0$.

Our second assumption describes how the step size $\alpha$ should depend on $m$, $d$, and the SGD iteration $t$. Recall the nomenclature "fixed" and "decreasing" as qualifiers for step-size, which describe a dependency on $t$ (but not $d$). Recall that $\lambda$ is the regularisation parameter.

**Assumption 2 (a).** $\lim\limits_{m \to \infty} \alpha^{(t)} \lambda \sqrt{\frac{m}{d}} = 1$.

Assumption 2 (a) allows for fixed or decreasing step sizes, such as $\frac{1}{\lambda} \frac{\sqrt{d}}{\sqrt{m+r(t)}}$ for increasing but finite $r$. Assumption 2 (a) suffices for our limiting result to hold. If we want to additionally quantify the distance between the limit and finite-dimensional kernels, we use the stronger Assumption 2 (b), which in particular requires a fixed step-size. Both variants 2 (a) and 2 (b) result in a limiting step size of 0, under Assumption 1.

**Assumption 2 (b).** *We have a fixed step-size* $\alpha^{(t)} = \frac{1}{\lambda} \sqrt{\frac{d}{m}}$.

We will find that in our setup, the DEKER is a composite function involving NNK building blocks.

### 3.3 Error function inverse link and Gaussian likelihood to match a random mapping

We will find that the update rule $\mathsf{G}$ of the DEKER is a composite function involving NNK building blocks. In order to clearly highlight the role of these NNK building blocks, we first present a special case before presenting our more general Theorem 4. This provides a clear link between the statistical likelihood model and closed form expressions for the DEKER update rule.

We choose an exponential family, canonical nonlinearity and random mapping $\Gamma$. This particular setup leads to closed-form expressions for the NNKs involved in the update rule. As an activation function, we choose the error function $\mathrm{erf}(z) = 2/\sqrt{\pi} \int_0^z e^{-v^2}\, dv$ (closely related to the Probit function), and rely on a closed-form NNK derived in Williams (1997),

$$\kappa_{\mathrm{erf}(\cdot/\sqrt{2})}(\Sigma_{11}, \Sigma_{22}, \Sigma_{12}) = \frac{2}{\pi} \sin^{-1} \frac{\Sigma_{12}}{\sqrt{(1 + \Sigma_{11})(1 + \Sigma_{22})}}. \tag{18}$$

We show here the statistical modelling choices and their corresponding effect on the DEKER update rule. In this special case, our main result (Theorem 4) implies Corollary 3, as proven in Appendix G.1. Recall from § 2.3, that the designer needs to choose a log partition function $A$, and a canonical nonlinearity $R$. We map input $X \in \mathbb{R}^l$ to data $Y \in \mathbb{R}^d$ through a random mapping $Y = \mathrm{erf}(\mathsf{W}X/\sqrt{2}) + Q$, where $\mathsf{W} \in \mathbb{R}^{d \times l}$ and $Q \in \mathbb{R}^d$ contain i.i.d. standard Gaussian elements. The distribution of $Y$ given $\mathrm{erf}(\mathsf{W}X/\sqrt{2})$ is conditionally Gaussian, with conditional expectation $\mathrm{erf}(\mathsf{W}X/\sqrt{2})$ having elements between $-1$ and $1$. We therefore choose a matching inverse link function, to represent the conditional expectation as a function of features $\psi_X$. The inverse link function is $s^{-1}(a) = \mathrm{erf}(a/\sqrt{2})$. The log partition function is $A(\eta) = \eta^2/2$ and the sufficient statistic is $T(y) = y$. Since the likelihood is Gaussian, the canonical nonlinearity $R$ and inverse link function $s^{-1}$ are the same, as shown in the first row of Table 1. In this particular case, the activations $\rho$ and $\sigma$ are shown in the third row of Table 1.

**Corollary 3.** *Suppose input $X$ is mapped to data $Y$ by* $Y = \mathrm{erf}(\mathsf{W}X/\sqrt{2}) + Q$, *where* $\mathrm{erf}$ *is the error function and* $\mathsf{W} \in \mathbb{R}^{d \times l}$ *and* $Q \in \mathbb{R}^d$ *contain i.i.d. standard Gaussian elements. Choose the log partition function* $A(\eta) = \eta^2/2$. *Choose the canonical nonlinearity* $R(a) = \mathrm{erf}(a/\sqrt{2})$, *or equivalently, choose the inverse link function to be* $s^{-1}(a) = \mathrm{erf}(a/\sqrt{2})$. *This implies that* $\rho(a) = 2p(a)$ *and* $\sigma(a) = 2p(a)\,\mathrm{erf}(a/\sqrt{2})$, *where $p$ is the pdf of the standard Gaussian. Then $\kappa_\rho$ and $\kappa_\sigma$ are given by*

$$\kappa_\rho(\Phi_{11}, \Phi_{22}, \Phi_{12}) = \frac{2}{\pi \sqrt{(1 + \Phi_{11})(1 + \Phi_{22}) - \Phi_{12}^2}},$$

$$\kappa_\sigma(\Phi_{11}, \Phi_{22}, \Phi_{12}) = \kappa_\rho(\Phi_{11}, \Phi_{22}, \Phi_{12})\kappa_{\mathrm{erf}(\cdot/\sqrt{2})}(F_{11}, F_{22}, F_{12}), \quad where \quad \mathsf{F} = \left(\Phi^{-1} + \mathsf{I}\right)^{-1}.$$

*Let $C_{ij} = \kappa_{\text{erf}(\cdot/\sqrt{2})}(X_i^\top X_i, X_j^\top X_j, X_i^\top X_j)$. Suppose Assumptions 1 and 2 (a) hold. Then applying SGD to objective equation 17, the update rule $\mathsf{G}$ equation 3 exists and can be decomposed into $G$ equation 5 satisfying*

$$G(\Phi_{ii}, \Phi_{jj}, \Phi_{ij}; X_i, X_j) = \frac{1}{\lambda^2}\left( C_{ij}\kappa_\rho(\Phi_{ii}, \Phi_{jj}, \Phi_{ij}) + \kappa_\sigma(\Phi_{ii}, \Phi_{jj}, \Phi_{ij}) \right).$$

Note that in this case the component $G$ of the update rule $\mathsf{G}$ can be computed entirely in closed form. The $2 \times 2$ matrix $\mathsf{F} = (\Phi^{-1} + \mathsf{I})^{-1}$ has a simple closed-form in terms of $\Phi$ (see Appendix G.1)[4]. Recall that the decomposition of $\mathsf{G}$ into $G$ says that, by plugging equation 5 into equation 3, for each $ij \in \{11, 22, 12\}$,

$$\Psi_{ij}^{(\tau+1)} = G\big(\Psi_{ii}^{(\tau)}, \Psi_{jj}^{(\tau)}, \Psi_{ij}^{(\tau)}; X_i, X_j\big). \qquad \text{That is,} \quad \mathbf{\Psi}^{(\tau+1)} = \mathsf{G}(\mathbf{\Psi}^{(\tau)}; X_1, X_2).$$

### 3.4 General case

We now consider the general setting, allowing for arbitrary $(A, s)$ pairs and random mappings $\Gamma$. In order to analyse this generalised setting, we require one additional definition equation 19 and two additional assumptions 3 and 4.

In the most general setting, the DEKER includes some non-symmetric (hence not PSD and not a kernel) cross terms. Given two activations $\zeta_1$ and $\zeta_2$,

$$\kappa_{\zeta_1, \zeta_2}(\Sigma_{11}, \Sigma_{22}, \Sigma_{12}) \triangleq \mathbb{E}_{(\chi_1, \chi_2)^\top \sim \mathcal{N}(\mathbf{0}, \mathbf{\Sigma})}\big[\zeta_1(\chi_1)\zeta_2(\chi_2)\big]. \tag{19}$$

The third assumption says that if the inner products were empirical estimates of an expectation, the resulting expectation is real valued and finite. Recall that $T$ is the sufficient statistic of the exponential family, $\Gamma$ is the random mapping from input space to data space, and $\sigma(a) = (A \circ R)'(a)$

**Assumption 3.** *The expectation $K(a) = \mathbb{E}_Z\left[\big(T(\Gamma(X)) \odot \rho(aZ) - \sigma(aZ)\big)^2\right]$ is finite for all $X \in \mathbb{X}$ and $a \in \mathbb{R}$, where $Z$ is a standard Gaussian random variable.*

The fourth assumption describes the properties of the random mapping $\Gamma : \mathbb{X} \to \mathbb{Y}^d$ as $d \to \infty$. In order to understand what happens to the solutions found by SGD as $d$ becomes large, we need the inputs which are passed through $\Gamma$ to be well-behaved. It suffices that a kernel and average defined by $\Gamma$ converges. We call the limiting kernel $c$ the *explicit kernel*, which contrasts with our implicitly defined DEKER. We give examples in Appendix F.

**Assumption 4.** *The PSD kernel $c$ defined by $c(X_1, X_2) \triangleq \lim_{m \to \infty} \frac{1}{d} T(\Gamma(X_1))^\top T(\Gamma(X_2)) = \mathbb{E}T(\Gamma(X_1))^\top T(\Gamma(X_2))$ is finite. Similarly, the mean function defined by $\mu(X_1) \triangleq \lim_{m \to \infty} \frac{1}{d} T(\Gamma(X_1))^\top 1 = \mathbb{E}T(\Gamma(X_1))^\top 1$ is finite.*

We are now ready to state our main result.

**Theorem 4.** *Suppose Assumptions 1, 2 (a), 3, and 4 hold. Let $C_{ij} = c(X_i, X_j)$ and $\mu_i = \mu(X_i)$ be as defined in Assumption 4. Then applying SGD to objective equation 17, the update rule $\mathsf{G}$ equation 3 exists and can be decomposed into $G$ equation 5 satisfying*

$$G(\Phi_{ii}, \Phi_{jj}, \Phi_{ij}; X_i, X_j)$$
$$= \frac{1}{\lambda^2}\left( C_{ij}\kappa_\rho(\Phi_{ii}, \Phi_{jj}, \Phi_{ij}) - \kappa_{\sigma, \rho}(\Phi_{ii}, \Phi_{jj}, \Phi_{ij})\mu_i - \kappa_{\rho, \sigma}(\Phi_{ii}, \Phi_{jj}, \Phi_{ij})\mu_j + \kappa_\sigma(\Phi_{ii}, \Phi_{jj}, \Phi_{ij}) \right).$$

*Here $\kappa_\sigma$, $\kappa_\rho$, $\kappa_{\sigma, \rho}$ and $\kappa_{\rho, \sigma}$ are as defined by equation 11, equation 19 and Proposition 2.*

---

[4]The matrix $\mathsf{F}$ arises because the activation functions $\rho$ and $\sigma$ involve the pdf of a standard normal distribution $p$ with covariance matrix $\mathsf{I}$. When $p$ is multiplied with the pdf of the bivariate normal distribution with covariance $\Phi$, this has an effect of computing harmonic means of covariances $\Phi$ and $\mathsf{I}$, resulting in $\mathsf{F}$.

**Proof sketch** Our main result is a constructive proof for the existence of an update rule $\mathsf{G}$, as posed in equation 3. In order to prove our main result, we will need to prove a series of lemmas, as detailed in Appendix E. The intuition behind these lemmas is as follows. The stochastic gradient $\frac{\partial}{\partial\phi}L(\phi; X, \mathsf{V})$ evaluated at an arbitrary point $\phi \in \psi$ for input $X$ and random $\mathsf{V}$ is the sum of the gradient of the negative log prior and the stochastic gradient of the negative log likelihood,

$$\frac{\partial}{\partial\phi}L(\phi; X, \mathsf{V}) = \underbrace{\sqrt{\frac{m}{d}}\lambda\phi}_{\text{Gradient of negative log prior}} \underbrace{-\frac{1}{d}\mathsf{V}^\top\big(T(\Gamma(X)) \odot \rho(\mathsf{V}\phi) - \sigma(\mathsf{V}\phi)\big)}_{\text{Stochastic gradient of log likelihood}}. \tag{20}$$

Assumption 2 (a) means that if the limit were allowed to be applied, the gradient of the negative log prior term in equation 20 multiplied by the step size would look like $\phi$. This means that the update of SGD would reduce to the stochastic gradient of the log likelihood. To derive the kernel update rule, we then examine the inner product of the stochastic gradient of the log likelihood. We first convert the inner product of the stochastic gradient of the log likelihood to an approximate form that is easier to deal with (Lemma 15). We then confirm that the kernel update only involves the inner product of the stochastic gradients of the log likelihood (Lemma 16). Finally, we show that the inner products of the approximate form converges to a closed form update rule $\mathsf{G}$ (Lemma 17). Assembling these lemmas together yields Theorem 4. The detailed proof is given in Appendix E.

Recall again that the decomposition of $\mathsf{G}$ into $G$ says that, by plugging equation 5 into equation 3, for each $ij \in \{11, 22, 12\}$,

$$\Psi_{ij}^{(\tau+1)} = G\big(\Psi_{ii}^{(\tau)}, \Psi_{jj}^{(\tau)}, \Psi_{ij}^{(\tau)}; X_i, X_j\big). \qquad \text{That is,} \quad \Psi^{(\tau+1)} = \mathsf{G}(\Psi^{(\tau)}; X_1, X_2).$$

Note the cross terms involving $\kappa_{\sigma,\rho}$ and $\mu$ in Theorem 4, which were not present in the special case of Corollary 3. These cross-terms arise from random mappings $\Gamma$ with an average element that is non-zero. In the case of Corollary 3, these cross-terms cancel out.

Theorem 4 implies a fixed point condition by Theorem 1, providing a positive answer for equation 4.

**Corollary 5.** *Suppose the same setting as Theorem 4. If $\mathsf{G}$ is a contraction mapping, then the DEKER converges to a unique fixed point as $t \to \infty$. That is, for each $ij \in \{11, 22, 12\}$,*

$$\Psi_{ij} = \frac{1}{\lambda^2}\Bigg(C_{ij}\kappa_\rho\big(\Psi_{ii}, \Psi_{jj}, \Psi_{ij}\big) - \kappa_{\sigma,\rho}\big(\Psi_{ii}, \Psi_{jj}, \Psi_{ij}\big)\mu_i - \kappa_{\rho,\sigma}\big(\Psi_{ii}, \Psi_{jj}, \Psi_{ij}\big)\mu_j + \kappa_\sigma\big(\Psi_{ii}, \Psi_{jj}, \Psi_{ij}\big)\Bigg). \tag{21}$$

Whether $\mathsf{G}$ is a contraction can be determined by a derivative test and an identity given in Theorem 19, as we demonstrate in § G.2. Note that even if a unique fixed point does not exist (which may be the case if $\mathsf{G}$ is not a contraction), one may still compute with finite-$t$ iterates of SGD via Theorem 4.

We may compute iterates of SGD in the limit via the update rule for any $\tau$ to obtain $\Psi_{ij}^{(\tau)}$, which is the naive fixed point algorithm applied to equation 21. Alternatively, we may compute the $\ell$DEKER by solving equation 21 for each $ij \in \{11, 22, 12\}$ using any other fixed point solver.

**Notable special cases and relation to NNK and NTK** Some further examples arising from special choices of $A$, $R$ and $\Gamma$ (inducing corresponding $\rho$, $\sigma$, $c$ and $\mu$) are discussed in Appendix G.2. We find that the linear Gaussian ($A(\eta) = \eta^2/2$, $R(a) = a$) results in a DEKER that is a scale multiple of $c$ (Appendix G.2.1). We can recover an NNK with activation $\sigma$ when $A$ and $R$ are allowed to be general and $C$ and $\mu$ are set to zero (Appendix G.2.2). This means that any NNK with activaiton $\sigma$ may be expressed as a special case of a DEKER. The setting we found useful for our experiments (§ 4) is a nonlinearly parameterised Gaussian ($A(\eta) = \eta^2/2$ and $R(a) = \text{ReLU}(a)$) with a first-order arc-cosine kernel for $c$ and a corresponding mean function $\mu$. This setting admits a closed-form update rule for $\mathsf{G}$. While this setting is distinct from the NTK with ReLU activations, it shares superficial similarities in that both updates involve repeated iterations of kernels $\kappa_\rho$ and $\kappa_\sigma$. See Appendices G.2.4 and G.2.5 for details.

### 3.5 Sensitivity

In order to quantify the rate at which the infinite-dimensional, infinite-iteration DEKER converges with respect to the dimension, we need the feature mapping to be well-behaved. Lipschitzness and boundedness allows concentration inequalities to be applied.

**Assumption 5.** *Suppose $\rho$ is bounded or Lipschitz. Suppose $\sigma$ is bounded or Lipschitz.*

We may quantify the degree to which the infinite-dimensional, infinite-iteration DEKER is an invariant of SGD. We define specific values of finite dimensional $\phi$ and $\phi'$ using the infinite dimensional kernel fixed point $\Psi_{11}$, $\Psi_{22}$ and $\Psi_{12}$. This definition will serve as a good approximation of an invariant.

**Definition 6.** *Define*

$$r_1 = \sqrt{\Psi_{11}} \left(1, 0, \ldots 0\right)^\top \in \mathbb{R}^m, \qquad r_2 = \sqrt{\Psi_{22}} \left(\cos\omega, \sin\omega, 0, \ldots 0\right)^\top \in \mathbb{R}^m,$$

*where $\cos\omega = \frac{\Psi_{12}}{\sqrt{\Psi_{11}\Psi_{22}}}$. Then $r_1^\top r_2 = \Psi_{12}$, $r_1^\top r_1 = \Psi_{11}$, and $r_2^\top r_2 = \Psi_{22}$.*

We bound the residual of the finite dimensional kernel evaluated at an initial guess that is the solution of the infinite $(m, d)$ system. When this bound is small, intuitively speaking, the limiting solution $\Psi$ is "almost" an invariant of the finite-dimensional system. By invariant of SGD, we mean that $\Psi$ does not change as iterates of SGD increase. In other words, if we intitialise SGD close to the solution $\Psi$, future iterates of kernels will stay close to the initialisation with high probability.

**Theorem 7.** *Suppose Assumptions 1, 2 (b), 3, 4 and 5 hold. Let initial guesses be $\psi_{X_1}^{(0)} = r_1$ and $\psi_{X_2}^{(0)} = r_2$ as in Definition 6. Then there exist constants $Q_2, Q_3, c_2, c_3 > 0$ such that for all $\delta > 0$, $\epsilon_2 > 0$ and $\varepsilon_2$,*

$$\mathbb{P}\Big(\big|\overline{\Psi}_{12}^{(1)} - \Psi_{12}\big| \leq \varepsilon_1 + \varepsilon_2\Big) \geq 1 - \delta_1 - \delta_2,$$

*where*

$$\varepsilon_1 = \frac{K + \epsilon_2}{\lambda^2}(2\epsilon_1 + \epsilon_1^2), \quad \delta_1 = 2\exp\big(-c_2 dM_2\big) + \exp\big(-m\delta^2/2\big) \quad and \quad \delta_2 = 2\exp\big(-dc_3 M_3\big)$$

*and $\epsilon_1 = \sqrt{\frac{d}{m}} + \delta$, $M_2 = \min\left\{\frac{\epsilon_2^2}{Q_2^2}, \frac{\epsilon_2}{Q_2}\right\}$ and $M_3 = \min\left\{\frac{\varepsilon_2^2}{Q_3^2}, \frac{\varepsilon_2}{Q_3}\right\}$ and $c_3 > 0$ is some absolute constant.*

The proof is given in Appendix E. The probability decays to 1 exponentially in the minimum of $m$ and $d$, where we recall that $d$ is the number of conditionally independent observations in the exponential family and $m$ is the dimensionality of the latent variable. The closeness $\varepsilon_1 + \varepsilon_2$ may be configured to be close to zero by choosing $\varepsilon_2$, $\delta$ and $\epsilon_2$ to be small, thereby trading off against constants resulting in a slower exponential decay of probabilities.

## 4 Experiments

Recall that the ffDEKER is defined for finite SGD iteration $\tau$ as an inner product of finite $m$-dimensional features. The DEKER is defined for finite SGD iteration $\tau$ as a limit as $m \to \infty$ of an inner product of $m$-dimensional features. The $\ell$DEKER is defined as a limit as SGD iteration $\tau \to \infty$ of the DEKER. Although the DEKER and $\ell$DEKER are defined in terms of infinite dimensional features, evaluations of the DEKER and $\ell$DEKER are scalar values and can be used to form matrices with a finite number of rows and columns. These matrices can be used in kernel methods to build predictive algorithms.

In our analysis in the previous section, we considered $2 \times 2$ kernel matrices. Such analysis extends to $N \times N$ kernel matrices, where each element of the $N \times N$ kernel matrices may be related to an element of a $2 \times 2$ kernel matrix. All our implementations are vectorised, so that they operate on $N \times N$ matrices.

### 4.1 Measuring finite-width effects

We empirically measure the similarity of (finite-$\tau$, finite-$d$) ffDEKER matrices and (infinite-$\tau$, infinite-$d$) $\ell$DEKER matrices using the centered variant (Cortes et al., 2012) of kernel alignment (Cristianini et al., 2001), abbreviated CKA, as $d$ increases. We vary $d$ between 5 and 500 in steps of 5 and choose $m = d^{3/2}$. For control, we also measure the CKA between the (finite-$\tau$, finite-$d$) ffDEKER and the squared exponential kernel (SEK). See Figure 2, and Appendix J.1 for full details on the experimental setup. As expected (Theorem 4), the CKA between the DEKER matrices becomes larger as $d$ and $m$ increase, but not between the SEK and finite DEKER.

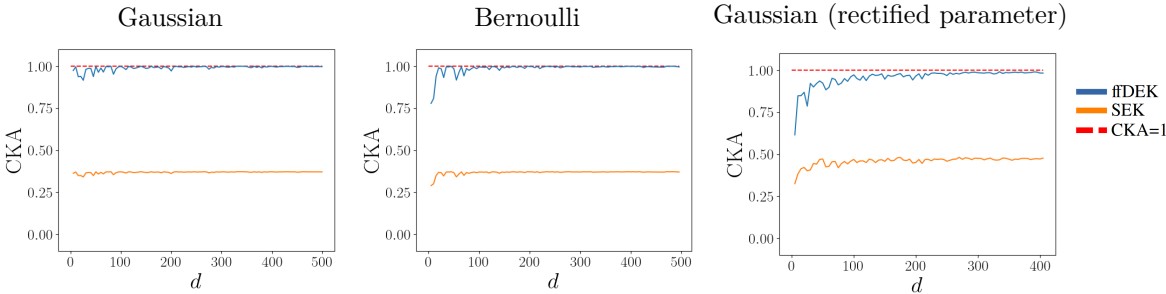

Figure 2: CKA between kernel matrices consisting of entries $\Psi_{ij}$ and $k_d^{(t)}(X_i, X_j)$ (Blue) and squared exponential kernel for control and $k_d^{(t)}(X_i, X_j)$ (Orange) for three choices of $A$ and $R$. (Left) Gaussian exponential family, $A(\eta) = \eta^2/2$ and $R(\eta) = \eta$. (Middle) Bernoulli exponential family, $A(\eta) = \log(1 + \exp \eta)$ and $R(\eta) = \eta$. (Right) Gaussian exponential family with rectified parameter, $A(\eta) = \eta^2$ and $R(\eta) = \text{ReLU}(\eta)$. The dashed red horizontal line at 1 represents the maximum value of CKA.

### 4.2 Inference using the DEKer

We use the DEKER for kernel ridge regression (Saunders, 1998) (KRR) (*cf* Gaussian process regression (Rasmussen & Williams, 2006)) on a suite of benchmarks. For each dataset, we first partition the data into an $80 - 20$ train-test split. Using the training set, we perform 5-fold cross-validation for hyperparameter selection using the default settings of sci-kit learn's `GridSearchCV`, which performs model selection based on the coefficient of determination. The hyperparameter grid we search over is described in Table 4, Appendix J.2. We then compute the RMSE on the held out test set using all training data. We repeat this procedure for 100 different random shuffles of dataset, and find the sample average and standard deviation RMSE over the random shuffles. The results are reported in Table 2. The input $X$ is preprocessed by subtracting the sample average and dividing by the sample standard deviation of each feature. Additionally, the target data $y$ is mean-centered and scaled by the sample standard deviation. The reported RMSE is after conversion of $y$ back to original units.

Since the DEKER is a strict generalisation of the NNK, we expect the DEKER to strictly out-perform the NNK. Any empirical performance result that may be obtained by previous investigations into the NNK with activations $\sigma$ (Lee et al., 2020) can be reproduced by a DEKER with the correct hyperparameter choice. We do not empirically explore uncertainty properties of corresponding Gaussian process models as others do (Adlam et al., 2020), but note that the same principle applies. We find that `GridSearchCV` sometimes picks out settings that correspond with an NNK, but often does not. The number of times `GridSearchCV` collapses the DEKER to the NNK is indicated in the last column of Table 2. Our results are consistent with the previously established observation that "NNKs frequently outperform NTKs" (Lee et al., 2020). More interestingly, we find that for each dataset, the DEKER performs as well or better than every other kernel, including the NNK.

| Data | DEKer or ℓDEKer | NNK | NTK | SEK | |
|---|---|---|---|---|---|
| yacht | **0.65 ± 0.21** | 2.13 ± 0.57 | 2.75 ± 0.58 | 3.62 ± 0.67 | 0 |
| diabetes | **54.51 ± 3.29** | **54.58 ± 3.30** | **55.05 ± 3.38** | **54.75 ± 3.32** | 70 |
| energy1 | **1.00 ± 0.11** | **1.01 ± 0.11** | 1.67 ± 0.14 | **1.08 ± 0.13** | 78 |
| energy2 | **1.58 ± 0.15** | **1.58 ± 0.15** | 2.10 ± 0.18 | **1.58 ± 0.16** | 68 |
| concrete | **4.94 ± 0.47** | **4.97 ± 0.47** | **5.05 ± 0.48** | 5.65 ± 0.39 | 60 |
| wine | **0.57 ± 0.02** | 0.61 ± 0.02 | **0.54 ± 0.02** | 0.62 ± 0.02 | 1 |

Table 2: RMSE of KRR models (± one standard deviation over 100 random seeds). We use the DEKer described in § G.2.5, which outperforms other kernels according to the sample average of the RMSE. Often the difference in performance is small compared with the standard deviation. The final column is the number of times the best DEKer found using `GridSearchCV` was an NNK. The datasets are UCI benchmarks — yacht (Gerritsma et al., 2013), diabetes (Kahn), energy (first and second value independently) (Tsanas & Xifara, 2012), concrete (Yeh, 2007) and wine (Cortez et al., 2009).

## 5    Conclusion

We introduced the DEKer, a kernel analogue of implicit neural network models. The DEKer is defined as the limiting inner product between two features computed using a feature update procedure as the dimensionality of the features goes to infinity.

We considered the problem of whether a deterministic update procedure for the DEKer exists (equation 3), and whether this update rule converges (equation 4). We focused on the special case where the features are latent variables in an exponential family PCA model (with not necessarily canonical link function) learnt using SGD. Leveraging the connection between infinitely wide explicit neural networks and kernel methods, we showed how in such a setting an explicit update rule can be computed. The update rule is a composition of functions involving NNK building blocks. While this type of update rule is not the one that is typically encountered in deep learning, which usually updates weights using SGD, it results in an interesting limiting model that shares connections to the NNK.

The DEKer has a number of interesting properties. The DEKer is able to recover instances of the NNK, and also resembles the NTK. Importantly, unlike the NNK and NTK, the deep layer structure of the DEKer is motivated entirely from an optimisation perspective. The activation functions (and thus kernels) involved in the computation of the DEKer can be related back to statistical modelling assumptions on the data through the exponential family. In particular, the activation functions share a connection to the log partition function and inverse link function of the exponential family. On a series of benchmarks, the DEKer performs as well as or outperforms the NNK, NTK and SEK.

Our work admits several natural extensions. The matrix $\mathsf{V}$ which represents a linear transformation or fully connected layer may be constrained to resemble a convolutional layer, and we expect a convolutional variant of the DEKer to be tractable (Novak et al., 2019). Since our construction is probabilistic, the Laplace approximation may yield a tractable means of obtaining principled uncertainty estimates for kernel methods beyond the regular Gaussian process framework. Since the DEKer satisfies a fixed point equation, implicit differentiation may be used to compute derivatives of the DEKer with respect to its hyperparameters, mirroring the neural network counterpart (Bai et al., 2019).

We considered the special case where the DEKer equation 2 is defined using features that are solutions to (latent variable model) optimisation problems using SGD. The resulting DEKer satisfies a fixed point equation. In future, other work might consider other types of problems and algorithms, resulting in DEKers which satisfy other types of conditions. Concurrent work (Cirone et al., 2023) considers solving neural controlled differential equations using Euler discretisations. Their resulting kernel satisfies a certain type of partial differential equation. We believe these two results, along with classical descriptions of Gaussian processes as solutions to stochastic differential equations (Rasmussen & Williams, 2006, Appendix B), might represent special instances of a more general framework.

We hope that our optimisation view and deterministic kernel update rule stimulates new research in both deep learning and kernel methods.

## Acknowledgments

Both authors would like to acknowledge the support of CSIRO's Machine Learning and Artificial Intelligence Future Science Platform.

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
