# OpenReview forum: "Stochastic gradient updates yield deep equilibrium kernels"
_TMLR — Accepted by TMLR_

### Review · Reviewer_1C9V · 2023-02-28

**Summary Of Contributions:**

This paper studies a family of recursively defined kernels (called DEK, deep equilibrium kernels) that are associated with specific stochastic updates on the features. More precisely, the authors defined a sequence of features $\phi^t$, where $\phi^{t+1}$ is obtained from $\phi_t$ via an SGD update which maximizes the log-likelihood associated with a certain stochastic exponential family. Under several assumptions and in certain scaling limits, the authors prove convergences of the kernel $\phi_t^T \phi_t \to K_t$ and obtain the recursive formula  $K_{t+1} = G(K_t)$ for some $G$. If $G$ is a contraction, then $K_t\to K^*$ for some fixed point.

The authors point out that the DEK is different from the NNK (Lee 2018) and compare performance between them on some benchmarks (UCI datasets?) and see some benefits from DEK.

**Audience:**

Yes

**Claims And Evidence:**

Yes

**Requested Changes:**

1. Streamlining the presentation. Exponential family/link functions, NNK, are splashed around the paper. I am trying to figure out what are the main results of the paper, and it is only evident after reading the first part of section E in the appendix. What you need is just a stochastic update to the features, while the (stochastic) exponential family is one approach to generate such an update. As such, this seems to be a particular setting in the Tensor Program (Yang, [2])

2. Two major points regarding the update rules. 1. SGD update is to the features (vs. variables) with random objectives $\Gamma(X)$. 2. $V^t$ are freshly resampled (vs. sharing) during each update. Is there an explanation behind doing so? Or is it just a technical consideration to make sure the arguments can go through? Please motivate the update rule (15).

3. There is a large knowledge gap between update (15) and Theorem 4. I only understand the connection and main result of the paper after reading its proof in Section 4. Please fill in the gap and move some parts of the content in Section E to the main text. The idea is not difficult, but the presentation makes it hard to follow.


4. Too many notations without proper explanation. Several examples are in order. (1) I think average readers will have a hard time to understand the notations in section 1.1, which are not clearly defined. (2) What is $U$ in the second paragraph of page 4, which is later defined. (3) What is $L$ in section 3.2? which is defined much later in eq (13). (4) The inner product $\psi \times \psi \to \psi$ above Section 3 is problematic, the same as the definition of $\kappa$ (the domain is incorrect). $\psi$ a subset in $\mathbb R^m$. (5) I don't understand what "over-parameterised" and "under-regularised" mean on page 9. (6) $W$ is used for both vectors in matrices



others.
- NTK is not very relevant to the paper, or am I missing something?
- In the appendix E, why the norm of $\|\psi_X\| $ is bounded, isn't it the size is of $m$, which goes to $\infty$ by the assumption you made.


[2] Tensor Programs I: Wide Feedforward or Recurrent Neural Networks of Any Architecture are Gaussian Processes
 https://arxiv.org/abs/1910.12478

**Strengths And Weaknesses:**

# Strength

1. The family of kernels studied are quite interesting as well as their connection to stochastic exponential family, though I have some concern about this setup.

2. The paper covers a variety topics in ML, e.g. Neural networks kernels, deep equilibrium models, exponential family, etc.

# Weaknesses.

1. The paper is hard to follow. The paper requires some major changes to improve readability. See details comments in the next section.

2. Several settings in the paper seem unnatural, e.g., SGD updates on features (vs. parameters), optimization objective is stochastic from step to step, etc.

3. Connection to deep equilibrium models (Bai) is unclear. Is it the infinite-width limit of DEQs (this is what I expect from the name DEK)? There is a key difference regarding weight-tying, please clarify. In addition, what is special about `equilibrium` part of the DEK other than being a fixed point of certain iteration? E.g., one can also iterate the NNKs (Lee) to obtain a fixed point.

4. Experiments need some clarification. What is the dataset's name, UCI? Can you also compare it with existing works, e.g. table 2 in [1]. In addition, it will be good to have some comparison vs Lee (and potentially follow-up paper) on other dataset, e.g. CIFAR-10.

5. The math/notations are difficult to follow. Many of them are left unexplained; see next section.

[1] EXPLORING THE UNCERTAINTY PROPERTIES OF NEURAL NETWORKS’ IMPLICIT PRIORS IN THE INFINITEWIDTH LIMIT

---

> ### Author Response · Authors · 2023-04-17
> **Thanks Reviewer 1C9V**
>
> Thanks for your thoughtful review.
> We are happy to hear that you found the family of kernels and their connection with exponential families interesting.
> We hope to address your concerns below.

---

> > ### Author Response · Authors · 2023-04-17
> > **Thanks Reviewer 1C9V**
> >
> > 1. **Streamlining presentation.**
> > We will move the first part of Appendix E to the main text. The results of [2] can unfortunately not be applied to obtain our results. Applying the result of [2] only proves a limiting results, whereas our theorem 8 and analysis building up to theorem 8 describe finite width effects in terms of concentration inequalities. We have added a discussion of this point to our revision.
> > Additionally, we think that many readers will appreciate that our proofs are self contained and do not rely on invoking a tensor program.
> > This makes it easier to see the connection between activation functions and the exponential family, should the reader wish to dig into the proofs.
> > Also note that while the results of [2] could be used to prove some of our result, the connection between latent variable models and neural network kernel models is to the best of our knowledge entirely new.
> > Please also see response 1. to Reviewer xp66.
> >
> > 2. **The update rule.** You are correct in saying that this is a technical consideration to make sure the arguments go through. The optimisation objective $\mathcal{L}$ is deterministic across all steps.
> > We apply SGD to the optimistion objective by considering stochastic gradients $\nabla L$ of $\mathcal{L}$ (as is typical).
> > As you point out, our the SGD updates happen on features and not parameters (as we mention following equation (1)).
> > We agree that this is not typical.
> > However, we are motivated to study this setting to answer the question of ``what happens if updates \emph{were} carried out in feature space instead of parameter space?'' Also, we believe our resulting limiting model is interesting enough in its connection to neural network kernel models to sufficiently motivate updating features. In other words, we choose to analyse feature updates because they result in an interesting model that shares similarity but is distinct from existing models. Another future work might consider jointly updating features and parameters. This would be a sort of combination of the setting of the NTK and the DEK. Beyond this, our setup is one that allows for the technical arguments can go through.
> >
> > 3. **Presentation:**. Please see 1.
> >
> > 4. **Notations.** $\mathsf{U}$ is a suitably general parameter object. As you suggest, we will move its definition so that it is now in the same sentence as it is first introduced.
> > Thanks for spotting the typo around the inner product, which we will fix.
> > We will add a description of why we call the setup over-parameterised and under-regularised close to where you referred to it.
> > Over-parameterised is first mentioned at the top of page 3, where it is intuitively meaning the number of parameters grows much faster than the amount of data. Under-regularised here means that the strength of the regularisation coefficient is smaller than what it would be with a typically scaled Gaussian prior. We will make sure to discuss these points further where you refer to them. We will mention in the notation section that we use sans serif fonts to denote matrices and serif fonts to denote vectors. So $W$ is a vector and $\mathsf{W}$ is a matrix.
> >
> > **others:**
> > - The NTK is a model that shares structural similarity with the DEK, but is not related beyond that. See also the response to Reviewer xp66's second point.
> >
> > - Regarding the norm, are you referring to for example the analysis occurring around the bottom of page 28 and beginning of page 29?
> > Assumption 3 (invoked in Lemma 16 and 17) says that the right hand side (i.e. top of page 29) is finite, since it converges to a quantity bounded in terms of the $K(a)$ defined in Assumption 3 by a law of large numbers. Therefore the required norm is finite.
> >
> > - **Connection to DEQs:** We will add an updated discussion in the paper. Our construct is not the infinite-width limit of DEQs (the second paragraph on page 3 is somewhat confusing on this, we will make sure to remove this ambiguity in the updated version). Rather, it is a way of constructing a kernel which satisfies a fixed point equation via a latent variable model optimisation problem solved using SGD. We indeed call our model equilibrium for no other reason than being a fixed point of a certain iteration. While [1] can also obtain a fixed point model, they do so by direct construction, whereas we do it via a more fundamental construction involving SGD iterates (i.e. SGD updates yield DEKs).
> >
> > - **Experiments**. The experiments are indeed UCI. We will add appropriate references in the revision. Regarding experimental comparison with existing works, we may obtain the results of [1] using a special case of a DEK (see response 2 to Reviewer xp66 and Appendix G.2.3). By using an appropriate special case of a DEK, we may recover an NNK used by [1] with their chosen activation function. Therefore, any empirical result obtained in [1] can be obtained by a DEK, with an appropriate hyperparameter choice. We will make this discussion clear in the revision.

---

### Review · Reviewer_RRbN · 2023-04-04

**Summary Of Contributions:**

This work proposes a new kernel method called deep equilibrium kernel, the kernel is shown to be a generalization of common kernels in deep learning

**Audience:**

Yes

**Claims And Evidence:**

No

**Requested Changes:**


See weakness. I think extensive discussion and empirical validation of how DEK explains deep equilibrium models is needed

**Strengths And Weaknesses:**

This works introduces a kernel for the deep equilibrium models. Empirically, the kernel is shown to outperform the NTK kernel


Weakness:
The motivation and the promise of the paper is to propose a DEQ kernel to understand deep equilibrium models. However, I am not sure the question is answered throughout the paper: can the DEK understand deep equilibrium models? This should deserve extensive discussion and empirical validation. For example, I would like to see that a standard deep equilibrium model, as we increase its width, approaches the theoretical predictions of the DEK theory. This part is distinctively lacking in the current draft

---

> ### Author Response · Authors · 2023-04-17
> **Reviewer RRbN**
>
> It was **not** our intention to propose a DEQ kernel to understand deep equilibrium models.
> As we point out in the box at the bottom of page 2, our intention is to take an update rule in feature space (of a latent variable model), define a kernel in infinite dimensional feature space, and then analyse the update rule in kernel space. By analyse the update rule in kernel space, we mean find a mapping $\mathsf{G}$ which takes a kernel at iterate $t$ and outputs a kernel at iterate $t+1$, find whether $\mathsf{G}$ admits a closed form, and finally and discuss whether $\mathsf{G}$ converges to a fixed point. This is not building a DEQ kernel to understand deep equilibrium models. If you can point us to parts of the paper where we unintentionally gave you this impression, we will happily update these parts of the paper in the revision. We would not like to make any claims that are not supported by evidence. We will add an updated discussion in the contributions section, emphasising that our paper is not about building a DEQ kernel to understand deep equilibrium models.
>
> An analogue for the empirical validation of asymptotic theory approaching finite width models that you ask for (with the revised understanding of the purpose of the paper), is given in section 4.1. Here, we compare the kernel matrix of a finite-feature model with an infinite feature model in three different settings. We also compare the finite feature model with the (incorrect) squared-exponential kernel, as a control. We observe that our model faithfully describes finite-width models in all three settings, especially when $d$ is large.
>
> Thanks for your consideration, and we look forward to discussing any further concerns you have.

---

### Review · Reviewer_xp66 · 2023-04-13

**Summary Of Contributions:**

This paper considers a special case of implicit neural networks, where the inference(feedforward) through a layer is equivalent to a SGD update step of a certain optimization problem. In the infinite feature dimension limit, it tries to find a closed-form update rule $G$ of the DEK kernel, which corresponds to the SGD update step in the feature space. The theory shows that, under certain assumptions, an update rule $G$ exists and in certain speical cases $G$ has closed form expression.

**Audience:**

Yes

**Claims And Evidence:**

Yes

**Requested Changes:**

1. The problem should be setup clearly early on, and physical meaning of each quantity should also be clarified. It is better to give an illustrating example.

2. Clarify the connection between DEK and NNK, NTK.

3. Clarify the different meaning of SGD of this paper in the abstract and early in introduction.

4. Talk about the limitation (or extenstion) to general cases where the features are not obtained by SGD-like inference.

**Strengths And Weaknesses:**

**Strengths**

The most interesting result of the paper is that it finds the close-form expression for the DEK update, for several special settings of deep implicit neural network. For more general cases, it also provide an relation of $G$ with some quantities, which I believe can be numerically computed.

The proving techniques looks novel and contains new insights. It may be helpful for solving similar problems in other settings.

**weakness**

The current results are limited to the cases where the inference is done via SGD-like operation. I think it will be more interesting if it can apply to more standard settings.

I don't quite see the connection between the DEK with the kernels arising from neural networks, such as NNK, NTK. The paper spends a large space to describe those kernels, however, none of them seems to be DEK.

The paper is potentially misleading when talking about SGD, especially in the title, abstract and introduction. Because SGD often referes to the training of a neural network; however, in this paper SGD referes to the inference through a layer. The paper should clarify this difference early on in the paper.

The problem is not clearly setup until very late in the context. For example, in section 1.1 when talking about feature space, it is not clear what the feature should be, and what the dimension m means. Still in section 1.1, it is not clearly explained why one should consider the 2x2 matrix, instead of a nxn matrix. In addition, in section 3.2, it is not clear what the random object V is. I think the problem should be setup clearly early on, and physical meaning of each quantity should also be clarified.

It is confusing to me that, in section 2.4, the inner product of two features is a feature "$\psi \times \psi \to \psi$". Is that a typo?

---

> ### Author Response · Authors · 2023-04-17
> **Thanks to Reviewer xp66**
>
> We are glad that you found the closed-form expression for the DEK update interesting. We share your belief that the proving techniques might contains new insights, and hope they will be helpful for solving similar problems in other settings.
>
> **Requested changes:**
>
> 0. Thanks to you (and also Reviewer 1C9V) for pointing out the typo for the target space of the inner product. This should have been typeset in a different font, and this bug was introduced during a find and replace operation. The revision will fix this typo.
>
> 1. **Problem setup and presentation:**
> Reviewer 1C9V holds a similar concern. In order to allay your concerns, we propose a number of changes. We will forward reference to objects in section 3.2 when discussing them in section 1.1. We will elaborate on what we mean in section 1.1 on the currently terse explanations (e.g., instead of $\psi_X$ being the 'solution to the problem', it should be more concretely the solution to the point estimate of a latent variable model obtained via the optimsation problem (17). We will mention that it suffices to mathematically consider the $2 \times $ matrix, because every element in the $N \times N$ matrix can be related to the element of a $2 \times $ matrix. In implementation, we of course use an $N \times N$ matrix for efficient batching. We will mention this again in the experiments section. When we first mention $\mathsf{V}$ in section 3.2, we will call it a 'random linear transformation' instead, and forward-reference to the prior we place over $\mathsf{V}$ in the subsequent paragraphs.
> We will add a new subsection at the end of section 1, which discusses the example in corollary 3.3 in terms of the physical objects.
> We will move the first part of Appendix E to just before section 3.3 to give an intuition of the proof and the result.
>
> 2. **Clarify the connection between DEK and NNK, NTK.**
> We decided to devote roughy 1 page of the background section to the NNK because our main results are expressed in terms of the NNK integrals (11) and (12). These integrals are the functions denoted by various annotations of $\kappa$. We additionally added an extra paragraph on the NTK, because while it is not directly related to our results, it anticipates reader confusion about the relationship between NNK and NTK. We will add a further paragraph to section 2 which discusses the relationship between our DEK and the NNK and NTK. Referring to Appendix G.2.3, we will be clear to point out that any (multi-layer) NNK with activation function $\sigma$ is a special case of a DEK.
>
> 3. **SGD updates features:**
> We will make sure to include early on in the paper a discussion and emphasis that in our setting, SGD is performing inference (i.e. updating features, not parameters or weights). This is unlike a typical setting like the NTK, where weight and parameter updates are performed for a fixed network depth. The result of these differences is that in the infinite width limit, we **derive** depth as the result of the number of iterations of SGD.
>
> 4. **Other (non-SGD) features:**
> We agree that our work is limited to analysing features obtained using SGD. Since submission, we have been made aware of another work (likely to be accepted at ICML) which is similar in spirit to your comment. The authors consider a setup similar to ours, but instead of optimisation problems they consider differential equations, and instead of an SGD solver they use an Euler discretisation. Their resulting feature model is a kernel method/Gaussian process whose covariance function satisfies a differential equation. A unified view of solving problems (optimisation, differential equations, ...) with algorithms (SGD, Euler's method, ...) to define kernels via infinite dimensional features that satisfy a problem (fixed point equation, differential equation) is one we find very interesting and exciting, and we hope that research into this topic is done in future. We believe our result will be a useful special case step to building such a general extension. We will add a discussion to this point in the revision, and also cite the ICML paper if it is accepted.

---

### Review · Reviewer_NhqZ · 2023-04-27

**Summary Of Contributions:**

This paper analyzes deep equilibrium models in the kernel regime, taking the dimensionality of the implicit representations to infinity. They define kernels in terms of the features of an implicit architecture, that are the approximate fixpoints resulting from  $t$ iterations of naive forward iteration. They consider three variants of the Deep Equilibrium Kernel (DEK): 1) a kernel for finite-dimensional features (with a finite number of fixpoint solver iterations); 2) a kernel for infinite-dimensional features (with a finite number of fixpoint solver iterations); and 3) a kernel for infinite-dimensional features and infinite iterations of the fixpoint solver.

They provide four pages of background on fixed point theory, exponential families, and kernels for neural networks (e.g., NTK). Their main contribution is theoretical: they show that there exists a closed-form update rule in kernel space (e.g., operating on a $2 \times 2$ matrix of kernel evaluations) and that under certain conditions, this update converges to a fixpoint.

The authors perform a couple small-scale experiments, using the proposed DEK for kernel ridge regression. They show that, because DEQ is a generalization of the neural network kernel (NNK), it performs at least as well in all cases.

Unfortunately, there are several issues with the exposition. The writing could use another pass to improve clarity, as there are many places where the structure is disorganized and confusing to the reader, discussed in the Cons section below.

**Audience:**

Yes

**Broader Impact Concerns:**

None.

**Claims And Evidence:**

Yes

**Requested Changes:**

I'd mostly request changes around writing, presentation, and positioning of contributions as stated in detail in my review above.

**Strengths And Weaknesses:**

**Pros**
* The paper addresses an interesting topic, considering infinite-width, infinite-depth models (e.g., Deep Equilibrium Models with infinite feature dimensionality).
* The theory appears sound, although I did not check the proofs in detail.
* The mathematical exposition is mostly well-written, except for some notational issues mentioned below.
* Following Tsuchida & Ong (2022), they discuss some interesting connections between DEQ architectures and statistical concepts (e.g., the relationship between inverse link functions in statistics and activation functions in ML).

**Cons**
* In the 4th paragraph of the Introduction, what does "model-based deep learning" mean?
* I think it would be important to state the high-level contributions and takeaways before the start of Section 1.1, which already starts introducing fairly heavy mathematical notation.

* The exposition in the "Updates in feature space" paragraph of Section 1.1 is not clear.
* In Section 1.1, "$\psi_{X_1}$ is the evaluation of an implicit function of $X_1$" --> why not just "a function of"?
* "Let $\psi_{X_1}^{(t)}$ be a representation of the solution at iterate $t$." -> Iterate $t$ of what? The wording here seems backwards, because it defines the function $g^{(t)}$ _after_ $\psi_{X_1}^{(t)}$ when it should be defined before.
* It is confusing to introduce $X_1$ and $X_2$ in Eq. 1 as though they are central to the definition of $\psi_X^{(t)}$.
* Regarding Eq. 2, the paper should re-state that $m$ is the feature dimensionality, otherwise it is quite unclear, as $m$ does not appear in the right-hand side of any of the equations. The feature dimension is only briefly implied in the previous paragraph as $\psi_{X_1} \in \boldsymbol{\psi} \subseteq \mathbb{R}^m$.
* The writing in the "Deep equilibrium kernels" paragraph in Section 1.1 is unclear: it does not state why we would want to construct the matrix $\Psi^{(t+1)}$ or how it would be useful. Also, it would be clearer to write out the full matrix as
$$
\Psi^{(t+1)} =
\begin{bmatrix}
\Psi_{11}^{(t+1)} & \Psi_{12}^{(t+1)} \\
\Psi_{21}^{(t+1)} & \Psi_{22}^{(t+1)}
\end{bmatrix}
$$
  rather than writing "a PSD matrix containing $\Psi_{11}^{(t+1)}$, $\Psi_{12}^{(t+1)}$, $\Psi_{22}^{(t+1)}$.
* In the "Updates in kernel space" box: 1) clarify what the updates correspond to; 2) the writing is confusing, as $G$ is not described when it is first used.
* There is new notation $\Phi$ used without sufficient explanation.
* In the Contributions paragraph in Section 1.1, there is far too much referencing of future sections, equations, and theorems that have not been introduced yet, and this makes it hard for the reader to follow.
- "The objective (17) to which we apply SGD" --> This references an equation that is first defined 6 pages away, which the reader should not be expected to flip back-and-forth through the whole paper to find.
- The statement "Theorem 4 and Corollary 5 (informal)" is a bit jarring and out of place, as it is stated before any of the other theorems; if anything, I think that this writing can be simply stated in a paragraph without the Theorem block. The Theorem 4 statement again references Eq. 17, so this would need to be introduced in Section 1.1.
- The last sentence of Section 1.1, "We further quantify the degree to which the $\ell$DEK is an invariant of SGD when treated as an ffDEK." is not clear without context, and does not currently fit well in the introduction.
- "DEQ whose iterates are computed with stochastic approximation rather than as a deterministic fixed point iteration." --> This needs further explanation: what is mean by "stochastic approximation" here?
- In Section 2.1, the "unsupervised learning problem for a DEQ" is introduced, but the vast majority of work on DEQs considers the supervised setting. The work by Tsuchida & Ong 2022 should be cited here as one of the few that considers the unsupervised setting.
- I think that Section 2.4 on notation sould be moved earlier, near the introduction.
- In Section 2.4, it states that "superscripts indicate a layer or iteration of a naive fixed point solver, both of which turn out to be the same in our constructions" --> this makes it sound like it is unique to this paper's construction, while it is a general property of DEQs.
- "$V \phi$ having components with variance which stays in $d$ and $m$" --> what does it mean that the variance styas in $d$ and $m$?
- In Section 3.2, it is not clearly stated what "the dimension of the exponential family" means.
- In Section 3.2, it should be clarified whether the "random object $V$" refers to the model parameters?
- The presentation in Section 3.2 is unclear because it is not stated clearly enough that $m$ is the feature space dimensionality and $d$ is the dimensionality of the exponential family.
- In Section 3.3, it is unclear to mention corollaries and theorems that have not been stated yet, e.g., "In this special case, our main result (Theorem 4) implies Corollary 3."
- Structurally, it is _very_ confusing that Assumption 2(a) is stated in Section 3.2, while Assumption 2(b) is stated in Section 3.5---Assumptions 3 and 4 are listed in between them! What is the justification for this?
- In Corollary 3, where does $F$ come from? Why is the identity matrix added in $F$? Also, it is probably worth re-iterating that $\Phi$ is a matrix of kernel evaluations.
- There needs to be more discussion about the takeaways and implications of Theorem 8.
* It would be important to clearly define what is meant by "an invariant of SGD."
* The right-hand side of Eq. 20 is identical to that in Theorem 4, but is hard to parse and compare. It may be better to write $\Psi_{ij} = G(...)$  and then optionally write out the RHS.
* In Section 3.5, it is unclear what the sensitivity refers to, e.g., sensitivity to what?
* The subplots in Figure 2 are too small, and don't have legends (the legend information should not be just in the caption). In addition, each subplot should have a title to differentiate them, e.g., "Gaussian Exponential Family," "Bernoulli Exponential Family" and "Rectified Gaussian Exponential Family." Also, it is never stated what the dashed red line at $y=1$ corresponds to.

* This paper does not cite other work on neural tangent kernels for DEQ models, such as [1].

[1] Feng & Kolter. "On the Neural Tangent Kernel of Equilibrium Models" 2021.

**Minor**
* The abbreviation for Deep Equilibrium Kernel (DEK) would be pronounced identically to DEQ, which is the standard terminology for deep equilibrium models; the authors might want to consider an alternative abbreviation.
* Why are the equations in Eq. 2 expressed in terms of $t+1$ rather than $t$?
* A reference should be provided for Banach's fixed point theorem (Theorem 1).
* A great deal of space in the paper is devoted to background, comprising 4 pages (page 2.5 to page 6.5).
* Throughout the paper, it would be better to use the notation "Eq. 10" rather than just "(10)." This is especially confusing in Corollary 3, where it states "$G$ (3) exists and can be decomposed into $G$ (5) satisfying," because it is not obvious whether these parentheses are equation references or function applications.
* $U$ needs to be defined when it is first used, in $f_U(Z^*) = Z^*$.
* Extra space after "(NTK) (Jacot et al., 2018) ." on page 6.
* In Section 3.2, the reference for SGD is weird (Wright & Recht, 2022). Shouldn't this be Robbins & Monro, "A Stochastic Approximation Method" 1951?
* In Eq. 15, it is slightly confusing to write the same SGD update twice for $X_1$ and $X_2$.
* The first sentence of Section 3.3 is almost identical to the last sentence of Section 3.2.
* I don't know why Remark 6 is delineated as a Remark rather than just being part of the text.

---

> ### Author Response · Authors · 2023-05-04
> **Thanks for your review**
>
> We are glad that you found our paper interesting, and are very grateful for your in-depth review. Thanks for catching all the ambiguities, typos and general improvements. We appreciate that the writing could have used another pass to improve clarity. We attempt to address your concerns below; please let us know if you think any of our responses are unsatisfactory. We have uploaded an updated manuscript, with changes in blue. We are happy to follow up on any additional concerns you might have.
>
> - Model-based deep learning In order to properly describe this, we propose adding some extra words after the sentence which uses "model-based". Together with the previous paragraph, we believe this properly describes model-based.
> - High-level takeaways before sec 1.1: We added a new paragraph before section 1.1. Please see updated manuscript.
> - Implicit function. We added a sentence explaining why implicit function. Please see updated manuscript.
> - Backwards update rule and representation. We will change the wording, so that instead we first talk about the zeroth iteration and then define the tth iteration in order. Please see updated manuscript.
> - Agreed about the double definition of $X_1$ and $X_2$. Please see updated manuscript.
> - We have added description of $m$ and $d$ here and in a few extra places. Please see updated manuscript.
> - We added some words about why we care about the matrix. Also wrote out the matrix elements. Please see updated manuscript.
> - Changed the ordering so that $G$ is not referred to until after it is defined. Please see updated manuscript.
> - We have removed forward references. Please see updated manuscript.
> - Removed the "informal" statement, and replaced with a paragraph. Please see updated manuscript.
> - Added an explanation of invariant. Please see updated manuscript.
> - Thanks for pointing out the typo "stochastic approximation". We have changed the wording to "...infinitely wide DEQ whose iterates are computed with stochastic mapping that is resampled at every iterate rather than as a deterministic fixed point iteration." Please see updated manuscript.
> - We added this citation. Please see updated manuscript.
> - Moved notation section to beginning of background section. Please see updated manuscript.
> - We did not intend to claim this was unique to our setting. We have added "..., as is consistent with other DEQ works". Please see updated manuscript.
> - This was a typo. We meant to say "parameter Vϕ having components with variance which stays constant in d and m". Please see updated manuscript.
> - We have added further descriptions and reminders of $d$ and $m$ throughout. Please see updated manuscript.
> - Yes, these are model parameters. Please see updated manuscript.
> - Moved Assumption 2b to immediately after Assumption 2a. We originally put Assumption 2b after the first main results section because it only applies to the second main results section. We believe both presentation styles to be acceptable, but are happy to defer to your judgement. Please see updated manuscript.
> - $\mathsf{F}$ is a harmonic mean of two covariance matrices. It arises whenever we take a product of two Gaussian pdf's (as in some other settings, such as inference in graphical models). We have added a footnote to describe this. Please see updated manuscript.
> - We added some extra words before and after Theorem 7, including a description of invariant. Please see updated manuscript.
> - Added legend and titles to Figure 2. Please see updated manuscript.
> - Added a citation for [1]. Please see updated manuscript.
> - We changed to abbreviation to DEKer. Please see updated manuscript.
> - Expressions have (t+1) on the LHS so that they end up with (t) on the RHS. This is so that everything is consistent with the update rule equation (1), which has t+1 on the LHS. This decision is arbitrary (we could express things in terms of t and t-1), but consistent.
> - Provided a reference for Banach's fixed point theorem. Please see updated manuscript.
> - Changed equation references to "Equation x" instead of (x). Please see updated manuscript.
> - Defined $\mathsf{U}$ where it is first used. Please see updated manuscript.
> - We used Wright and Recht because it is a contemporary reference. It was not our intention to claim that Wright and Recht discovered SGD (or for root-finding, stochastic approximation) before Robbins and Monro. We have added Robbins and Monro as well. Please see updated manuscript.
> - Removed the remark status of Remark 6. Please see updated manuscript.

---

### Author Response · Authors · 2023-04-17
**Thanks to the reviewers**

We thank the reviewers for their informative reviews. We respond individually to reviews below with our proposed changes. We look forward to clarifying any further concerns that the reviewers have during the interaction and discussion period. If the reviewers feel that the proposed changes are not satisfactory, we encourage them to bring this to our attention.

---

### Decision · Action_Editors · 2023-06-01

**Recommendation:** Accept with minor revision

**Comment:**

Various concerns were raised regarding presentation and positioning.  While the authors have addressed many of them, some of the reviewers feel that more profound changes are required.  The authors are asked to take this feedback into account and appropriately update the text towards publication.

**Audience:**

Reviewer's generally agreed that the topic of the submission is of interest to the machine learning community.  Concerns were raised regarding the analyzed setting (inference via SGD updates) being less common in practice, but this was not seen as an impediment for publication by the majority of the committee.

**Claims And Evidence:**

Reviewers did not identify substantial gaps in terms of evidence supporting the submission's claims.